# Development of Ebola virus disease prediction scores: Screening tools for Ebola suspects at the triage-point during an outbreak

**Antoine Oloma Tshomba**[1,2]*, **Daniel-Ricky Mukadi-Bamuleka**[2,3], **Anja De Weggheleire**[4], **Olivier M. Tshiani**[2,3], **Richard O. Kitenge**[5], **Charles T. Kayembe**[6], **Bart K. M. Jacobs**[4], **Lutgarde Lynen**[4], **Placide Mbala-Kingebeni**[2,3], **Jean-Jacques Muyembe-Tamfum**[2,3], **Steve Ahuka-Mundeke**[2,3], **Dieudonné N. Mumba**[2,7], **Désiré D. Tshala-Katumbay**[2,8,9], **Sabue Mulangu**[2,3]

**1** Department of Public Health, University of Kisangani, Kisangani, Democratic Republic of Congo, **2** National Institute for Biomedical Research, Kinshasa, Democratic Republic of the Congo, **3** Department of Medical Biology, University of Kinshasa, Kinshasa, Democratic Republic of the Congo, **4** Department of Clinical Sciences, Institute of Tropical Medicine, Antwerp, Belgium, **5** National Emergency Program, Ministry of Health, Kinshasa, Democratic Republic of the Congo, **6** Department of Internal Medicine, University of Kisangani, Kisangani, Democratic Republic of the Congo, **7** Department of Tropical Medicine, University of Kinshasa, Kinshasa, Democratic Republic of the Congo, **8** Department of Neurology and School of Public Health, Oregon Health & Science University, Portland, Oregon, United States of America, **9** Department of Neurology, University of Kinshasa, Kinshasa, Democratic Republic of the Congo

* antotshomba@yahoo.fr

**Data Availability Statement:** All relevant data are in the manuscript and its Supporting Information files.

## Abstract

### Background

The control of Ebola virus disease (EVD) outbreaks relies on rapid diagnosis and prompt action, a daunting task in limited-resource contexts.

This study develops prediction scores that can help healthcare workers improve their decision-making at the triage-point of EVD suspect-cases during EVD outbreaks.

### Methods

We computed accuracy measurements of EVD predictors to assess their diagnosing ability compared with the reference standard GeneXpert® results, during the eastern DRC EVD outbreak. We developed predictive scores using the Spiegelhalter-Knill-Jones approach and constructed a clinical prediction score (CPS) and an extended clinical prediction score (ECPS). We plotted the receiver operating characteristic curves (ROCs), estimated the area under the ROC (AUROC) to assess the performance of scores, and computed net benefits (NB) to assess the clinical utility (decision-making ability) of the scores at a given cut-off. We performed decision curve analysis (DCA) to compare, at a range of threshold probabilities, prediction scores' decision-making ability and to quantify the number of unnecessary isolation.

### Results

The analysis was done on data from 10432 subjects, including 651 EVD cases. The EVD prevalence was 6.2% in the whole dataset, 14.8% in the subgroup of suspects who fitted the WHO Ebola case definition, and 3.2% for the set of suspects who did not fit this case

**Funding:** DDK-T received a from US National Health Institute to support the neuro-Ebola project (Grant reference NIH FIC/R01EY031894) NO - The funders had no role in study design, data collection, and analysis, decision to publish, or preparation of the manuscript.

**Competing interests:** The authors have declared that no competing interests exist.

definition. The WHO clinical definition yielded 61.6% sensitivity and 76.4% specificity. Fatigue, difficulty in swallowing, red eyes, gingival bleeding, hematemesis, confusion, hemoptysis, and a history of contact with an EVD case were predictors of EVD. The AUROC for ECPS was 0.88 (95%CI: 0.86–0.89), significantly greater than this for CPS, 0.71 (95%CI: 0.69–0.73) (p < 0.0001). At -1 point of score, the CPS yielded a sensitivity of 85.4% and specificity of 42.3%, and the ECPS yielded sensitivity of 78.8% and specificity of 81.4%. The diagnostic performance of the scores varied in the three disease contexts (the whole, fitting or not fitting the WHO case definition data sets). At 10% of threshold probability, e.g. in disease-adverse context, ECPS gave an NB of 0.033 and a net reduction of unnecessary isolation of 67.1%. Using ECPS as a joint approach to isolate EVD suspects reduces the number of unnecessary isolations by 65.7%.

## Conclusion

The scores developed in our study showed a good performance as EVD case predictors since their use improved the net benefit, i.e., their clinical utility. These rapid and low-cost tools can help in decision-making to isolate EVD-suspicious cases at the triage point during an outbreak. However, these tools still require external validation and cost-effectiveness evaluation before being used on a large scale.

## Introduction

Ebola virus disease (EVD) outbreaks have emerged in the tropical part of Africa with an increased incidence since the mid-nineties. The infection claimed tens of thousands of lives as its case fatality can reach ninety percent in the absence of any treatment [1–6]. Most EVD outbreaks have occurred in remote areas in which we note a lack of infrastructure and equipment, a few available diagnostic tools, and trained personnel. These conditions contribute to the further spread of the disease, compromising the outbreak control [6], and placing a heavy burden on the health systems and communities as well [7,8]. The control of EVD outbreaks relies on accurate and early detection of the pathogen to allow the rapid implementation of countermeasures [9]. Since the 2018 EVD outbreak [Equateur Province, Democratic Republic of the Congo (DRC)], the Cepheid® GeneXpert Ebola platform was used following the WHO's instructions as the reference standard for EVD diagnosis at the point of care. The GeneXpert® is a highly sensitive and specific, rapid, and fully automated dual-purpose quantitative RT-PCR that diagnoses EVD with a few skills and a short turn-around time [10]. Although on-site GeneXpert technology has shown beneficial results in controlling the EVD outbreaks [11], timely scaling-up of this technology face to an expanding or quickly changing outbreak context remains challenging because of the requirements such as electricity supply, laboratory infrastructure, personnel training, and the cost. In these contexts, and an absence of validated rapid diagnostic tests (RDTs), clinical signs and symptoms associated with epidemiological information can serve as an immediate tool to identify suspect-cases and to support patients' categorization and management. Early and accurate recognition of EVD can help improve the healthcare workers' (HCW) decision-making and epidemic control.

In the early stage of the infection, sudden fever onset and other non-specific symptoms, such as severe headache, muscle and joint pain, and fatigue characterize the EVD [12]. In tropical settings, many infectious diseases have a similar initial clinical presentation. EBOV-

specific symptoms appear at the last stage, further delaying the diagnostics in the absence of detection tools.

Often, HCWs must decide to isolate suspected cases without having any timely access to diagnostic tests. HCWs must weigh up the risk of sending true-positive EVD cases back to the community against the risk of admitting false-positive cases into the isolation wards. These errors in the triage result in the spread of the infection into the community or sustain nosocomial transmission in health settings, both delaying timely access to appropriate EVD care.

As for other epidemic pathologies, the clinical case definitions were developed, combining the signs and symptoms with risk factors (as having had contact with a confirmed case). The WHO clinical case definition (or local adaptation) is most commonly used through a recent meta-analysis showed its sub-optimal performance (pooled sensitivity of 81.5% (95% CI 74.1–87.2), and specificity 35.7% (28.5–43.6)) [13,14].

Thus, further research on simple clinical scoring systems is needed to add precision to the risk assessment in a single patient. Such simple tools, as for other conditions, are used, to better classify patients either for prognostic or management purposes [15–18].

Several prediction scores were developed to assess the individual risk of EVD using patient characteristics, clinical and epidemiological risk factors; and they were evaluated using accuracy metrics, e.g., sensitivity, specificity, and area under the curve [19–22]. However, the accuracy metrics only tell on how well a score discriminates EVD from non-EVD cases; but they do not assess the clinical usefulness of the score (i.e., their utility in clinical practice) by accounting for the clinical consequences of using them in healthcare practice. Consequently, none of the Ebola outbreak response teams has used these EVD risk scores during subsequent outbreaks. The score's clinical utility assessment provides the potential benefits and harms of using a score relative to the predicted absolute risk of disease to inform clinical decision-making in clinical practice.

This study aims 1) to develop an EVD clinical prediction rule using surveillance data from the 2018–2020 DRC EVD outbreak; 2) to evaluate the clinical usefulness of the prediction rule to improve healthcare workers' decision-making at the triage-point of EVD suspect-cased during outbreaks.

## Materials and methods

### Ethics statements

This study received the approval of the Ethics Committee of the University of Kisangani (Approbation No Ref: CER/005/GEAK/2017), and the authorization from the Secretariat Technique du Comité Multisectioriel de lutte contre la Maladie à Virus Ebola in DRC. This study used anonymized data from the surveillance and care unit collected as part of the EVD outbreak response (2018–2020). Since this study used de-identified data, the Ethics Committee did not consider individual patient consent as necessary.

### Study subjects, data collection, and reference standard for EVD confirmation

In this retrospective cohort study, we analyzed the epidemiological and clinical data collected routinely upon identification or admission of an EVD suspect case. We used the datasets of the surveillance and care units teams deployed in the areas covered by the EVD coordination team of Butembo, in the North-Kivu province. We included data with alive EVD suspects and the community deaths identified in the Butembo area between September 3, 2018 and February 17, 2020.

In this outbreak, the surveillance and clinical teams used a standard clinical case definition adapted from the WHO Integrated Disease Surveillance and Response (IDSR) guidelines to identify EVD suspect-cases [23].

The case definition contained notions of residence, signs, or symptoms without referring to contact information. The case definition was stated as follows: "*Any person alive or dead, living in North-Kivu, South-Kivu, or Ituri provinces or any person who traveled to these provinces during this period and who reported the following signs or symptoms; sudden fever onset and at least three of the following symptoms: vomiting, diarrhea, abdominal pain, conjunctivitis, rash, unexplained bleeding from any part of the body, muscle pain, intense fatigue, difficulty of swallowing, the difficulty of breathing, hiccups, or headache.*"

For each EVD suspect case, the surveillance and/or clinical team collected, guided by a standard notification form, data on socio-demographics (age, gender, residence, profession), putative epidemiologic routes of contagion (contacts, type of contact), date of symptom onset, and signs and symptoms at the time of diagnostic.

After the clinical and epidemiological assessment, each EVD suspect-case was subjected to the sample collection process (blood or oral swab) for confirmatory testing. While waiting for the result, EVD suspect patients were kept in the isolation wards within the triage center (TC), Ebola treatment center (ETC), or in the community for cases associated with refusal to be transferred to the care units.

For the Butembo areas, all samples from suspect-cases were tested in the EVD field laboratory set by the Institut National de Recherche Biomédicale (INRB) in Butembo. The confirmation of EBOV infection relied on the detection of EBOV nucleoprotein (NP) and/or glycoprotein (GP) in the clinical specimens through GeneXpert® (Cepheid, Sunnyvale, CA, USA) [10,11].

In our analysis, we used the GeneXpert® results as the reference for the EVD status. EVD confirmed case was defined as a suspect case with a GeneXpert® positive, i.e., EBOV Cycle threshold-values (Ct-values) $NP < 40$. A probable EVD case was defined as a suspect case with high-risk contact for whom a test was not available because no sample was collected. A non-case was a suspect case with a GeneXpert® negative result.

The surveillance team was led by the experts of the Ministry of Health (MoH), supported by the World Health Organization (WHO) and other partner teams. Ebola care units (Transit, treatment, and triage) were managed or supported by different teams such as MoH, WHO, Médecins Sans Frontières (MSF), the Alliance International for Medical Action (ALIMA), Samaritan Purse, etc. DRC Red Cross National Society) and research partners (INRB, Institute of Tropical Medicine of Antwerp, and US National Institute of Health) over time.

## Statistical analysis

**Data description.** Study data were available on Microsoft Excel 2016® sheets whereas they were analyzed with SPSS® IBM statistic version 20 software and R® package version 3.6.1(R Foundation for Statistical Computing, Vienna, Austria). Most clinical and epidemiological information consisted of binary variables. We dichotomized continuous measurements at the cut-off where the sum of specificity and sensitivity was maximized using the receiver operating characteristic curves (ROC). We tested associations between each possible predictor and EVD status using the chi-square test or Fisher exact test, as appropriate, at 5% of significance.

**Development of the scores.** Diagnostic accuracy, including the sensitivity, specificity, positive/negative likelihood ratios (LRs), and their 95% binomial confidence intervals for each demographic, clinical, and exposure information was calculated using 2x2 tables. For binary

variables, we considered the presence of a sign/symptom to be a positive test for the disease compared to the gold standard test. We selected associated predictors with a crude LR+ ≥2.0 or LR—≤0.5 and adjusted in multivariate logistic regression analysis according to an adaptation of the Spiegelhalter-Knill-Jones approach [24] as described in [25]. This approach combines the independent Bayes method and the logistic regression to construct a scoring system [26]. We selected the adjusted diagnostic predictors with aLR ≥1.5 or ≤ 0.67 to construct the scoring system. We obtained the diagnostic score for each retained predictor from the natural log-transformation of the aLRs, rounded to their nearest integer. Positive scores favored the presence of EVD, whereas negative scores argued against it. We obtained the total predictive score ($X$) for each suspect by summing the diagnostic score for each predictor. We assigned a value of 0 to the missing data. Finally, we computed from the total score for each suspect, the posterior probability of having EVD as follows:

$$P(D|\mathbf{X}) = \frac{\exp(\beta 0 + \sum \beta i * Xi)}{1 + \exp(\beta 0 + \sum \beta i * Xi)} \tag{1}$$

where $\mathbf{X}$ is the vector of all predictors $X_i$, D is the EVD status, $\beta_0$ is the constant of the regression equation and $\beta_i$ the coefficient of the $X_i$ predictor.

We developed two prediction models for the diagnosis of EVD: a clinical prediction model, the clinical prediction score (CPS), which included only clinical predictors, and an extended clinical prediction model, the extended clinical prediction score (ECPS), which included also demographic and epidemiologic predictors.

**Evaluation of the developed scores.** The performance of the developed prediction models (*i.e.* the extent to which each model distinguishes between EVD and non-EVD) was analyzed with a focus on their discrimination power, as described in [15,27]. Thus, we evaluated the discrimination performance of the CPS and ECPS by constructing their receiver operating characteristic curves (ROC) and calculating the areas under the curve (AUROC) with 95% confidence intervals (95%CIs) in the whole dataset and in subgroups of the EVD suspect's dataset (e.g., fitting or not the WHO case definition). An area of 1.0 indicates perfect discrimination power, whereas an area of 0.5 indicates no discrimination of the binary disease status (EVD versus non-EVD). Furthermore, we computed the diagnostic accuracy measures, e.g., sensitivity, specificity, PPV, and NPV, and their binomial 95%CIs at each cut-off point value of the score for each prediction model. Assuming that EVD prevalences in the two subgroups of suspects (e.g., those fitting or not fitting the WHO case definition) are different, we computed and described for each subgroup the diagnostic accuracy metrics to investigate the impact of the disease prevalence variation on the scores' accuracy performance.

Finally, we compared the AUROCs for prediction models developed by computing the critical ratio z as described in Hanley and McNeil [28]. We performed a 10-fold cross-validation on the logistic regression step and plotted calibration plots to evaluate the out-of-sample performance of the model with the selected predictors. Calibration refers to the agreement between predicted probabilities and observed proportions, or the actual risk of a clinical outcome. We used the Hosmer-Lemeshow goodness-of-fit test, and a model was considered statistically fit data if the p-value was greater than 0.1.

**Clinical usefulness of the prediction models.** To evaluate the clinical usefulness or utility of the developed prediction scores, we performed decision curve analysis, which depicts the net benefit (NB) of using a model at various ranges of threshold probabilities of interest, as described by Vickers and Elkin [29]. The net benefit defines the difference between the benefit for true cases (TP) who would receive correctly the intervention (isolation and care) based on the prediction of the model and the expected harm for non-cases (FP) who will erroneously

receive the intervention weighted by the odds of the patient's threshold probability. The model with the higher net benefit at a given threshold probability is the preferred model.

The net benefit (NB) for a given threshold probability was calculated as the following:

$$NB = \frac{\text{True positives}}{N} - \frac{\text{False positives}}{N} \times \frac{pt}{(1 - pt)} \tag{2}$$

In this equation, N denotes the total sample size and pt denotes the probability threshold above which a suspect is considered high-risk.

The weighting factor, the odds pt/(1-pt), captures the patient or health professional's values regarding the risks of false-negative and false-positive results.

Additionally, we plotted the decision curves for models and computed the reduction in the number of unnecessary isolation of suspects by using these models at a range of threshold probabilities of interest to compare our developed models. Additionally, the models were compared to two extreme strategies, isolating all suspects (where TP/N is the prevalence of EVD among the suspects and FP/N is one minus the prevalence of EVD in the NB formula) and isolating none of the suspects (where TP = zero and FP = zero, thus NB = zero at any threshold probability). For more detail on the technical approach of our derivation, please consider the paper by Vickers (http://www.decisioncurveanalysis.org/). In practice, implementing the decision curve analysis requires the following height steps:

1. Choose a value for $p_t$, e.g., the threshold probability, a composite of patient and health workers' preferences (a theoretical risk level where the expected benefit of isolating a suspect is equal to the expected benefit of avoiding it).

2. With this threshold probability as a cut-point, calculate the number of true and false positives using 2x2 tables that summarize the positive or negative results.

3. Compute the net benefit of the prediction model.

4. Repeat steps 2–3 by varying $p_t$ over a range of interest.

5. Plot the net benefit on the y-axis against $p_t$ on the x-axis to draw the decision curve for the model.

6. Repeat steps 1–5 for every model to compare.

7. Repeat steps 1–5 for the strategy that assumes all EVD suspects are positive.

8. Finally, a straight line drawn parallel to the x-axis at y = 0 represents the net benefit associated with the strategy of assuming that all suspects are negative.

We computed the reduction in the number of unnecessary isolations per 100 EVD suspects as described in [30].

$$\text{Unnecessary isolation per 100 suspects} = \frac{\text{NB of a model} - \text{NB of isolating all strategy}}{\frac{pt}{(1-pt)}} \times 100 \tag{3}$$

As a comparison, the EVD predictive model with the highest NB or the highest percentage of reduction of unnecessary isolation at a particular point of threshold probability will yield the best clinical usefulness at this point, e.g., more true-positive cases and, or fewer false-positive cases.

Finally, we evaluated our prediction models according to two additional clinical practice approaches in healthcare settings: joint and conditional tests or approaches. In both approaches, the suspects with no reported risk of exposure would be considered to not have

the disease and the clinical team would act accordingly. No additional action, e.g., isolation, would be required. In the joint approach, the clinical team should clinically examine all suspects at low-, intermediate-, and high-risk reported exposure and recommend for isolation only those with a predicted probability of EVD greater than 5% (the cut-off chosen to maximize sensitivity, about 90 percent, in disease adverse context). In the conditional approach, the clinical team should isolate all suspects with high-risk reported exposure irrespective of their predicted probability of the disease and then suspects at low and intermediate reported exposure having an EVD-predicted probability greater than 5%.

Thus, we first grouped EVD suspects into high-, intermediate-, and low-risk exposure according to their reported risk exposure to EVD. We define the categories of EVD risk exposure for suspects as follows: (1) no risk exposure if there has been no reported EVD risk exposure. (2) Low-risk exposure, any suspect who had direct contact with a living EVD patient who did not present diarrhea, vomiting, or bleeding at the moment of contact; or any suspect who had touched patient clothes or sheets which were not visibly soiled; or any funeral attendance or hospital visit but without reported direct contact. (3) Intermediate-risk exposure includes having had any contact with a living EVD patient who had diarrhea, vomiting, or bleeding, as well as any contact with clothes or sheets soiled with stool, vomit, or blood. (4) High-risk exposure: having cared for an EVD patient with diarrhea, vomiting, or bleeding; or having cleaned a patient room or clothes; or having handled patient waste; or any direct exposure to blood, body fluids (vomitus, urine, feces), or tissues from EVD patients; or direct skin contact with skin, blood, or body fluids from EVD patients; or having processed blood or body fluids from an EVD patient without protective equipment or direct contact with an EVD patient's dead body.

## Results

The surveillance system registered 10496 EVD suspect-cases during the outbreak in Butembo (DRC). Among them, 704 were confirmed-cases (including 673 confirmed EVD cases and 31 probable cases linked epidemiologically to a confirmed EVD case but no GeneXpert test done) and 9792 were classified as non-case cases during the epidemic. We excluded for analysis 31 suspects without laboratory results and those without clinical and epidemiological data (including 22 EVD cases and 11 non-EVD cases). Thus, we included for analyses 10432 subjects, among which 651 were EVD cases and 9781 were non-EVD cases as classified by the gold standard test used for screening during this outbreak. The EVD prevalence among Ebola-suspected people was 6.2% of the total sample. The disease prevalence was 14.8% among suspects who fitted the WHO case definition for the suspect (401 EVD cases in 2707 suspects) and 3.2% for those who did not fit (250 EVD cases in 7725 suspects). Fig 1 presents a flow diagram showing the number of EVD suspects included in the study and their outcomes (**Fig 1**).

The Table 1 summarizes the associations between the EVD status and the epidemiologic and clinical characteristics of the patients. This analysis revealed a significant association with the EVD status for all variables except only four predictors.

Table 2 details the accuracy metric results of epidemiologic and clinical predictors to predict the diagnosis of the EVD. Among the associated epidemiologically predictors, having reported any contact with EVD case had the highest sensitivity (65.4%, 95%CI: 61.5%–69.1%), followed by being more than 25 years old (61.7%, 95%CI: 57.8%–65.4%). For the clinically associated predictors, the highest sensitivity was 76.8% (95%CI: 73.3%–79.9%) for malaise and fatigue and the highest specificity was 99.9% (95%CI: 99.8%–99.9%) and 99.8% (95%CI: 99.7%–99.9%) for gingival bleeding and black-vomiting (vomito-negro), respectively. Additionally, the sensitivity for hemorrhagic signs was low, between 0.2% for hematuria and 11.7%

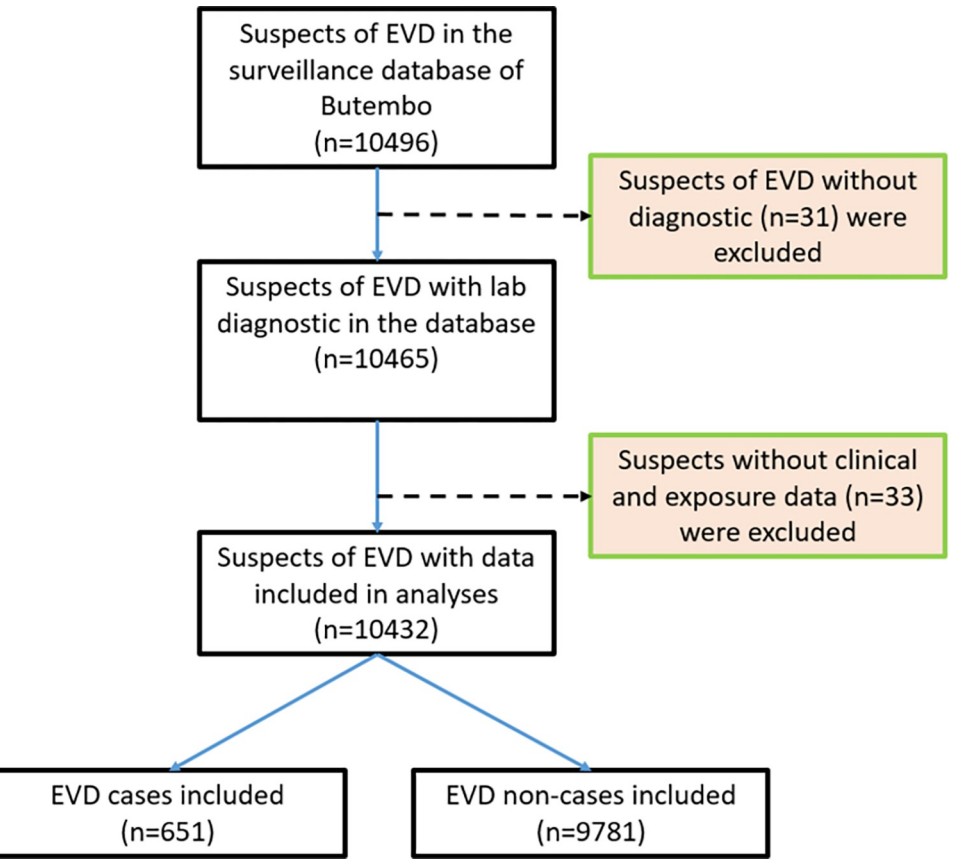

**Fig 1. Flow diagram showing the number of EVD suspects included in the study and their outcomes.** Classification of 10496 notified cases according to the WHO guideline case Definition for Ebola virus disease (EVD) during the Ebola virus Outbreak in BUTEMBO, RDC.

for any bleeding, and their specificity was relatively high, up to 100% for petechiae (Table 2). Further, most of these variables met the criteria for being the predictors of EVD diagnosis based on their positive and negative crude LRs as they are different from one for most of them. The highest positive LRs were 5.7(95%CI: 5.1–6.4) for any history of contact with an EVD case, 4.4(95%CI: 3.2–5.9) for eye redness, 45(95%CI: 15.5–64.0) for hematemesis, and 7.8 (95% CI: 4.5–12.0) for confusion-disorientation. The negative crude LRs were 0.4(95%CI: 0.3–0.4) for any history of contact with an EVD case among the epidemiologic predictors, and 0.5(95% CI: 0.4–0.6) for malaise/fatigue among the general clinical predictors. None of the hemorrhagic or neurological symptoms demonstrated the ability to rule out EVD infection in the analyses, as their negative LRs were equal to one or included one in their related confidence interval.

Based on their crude likelihood ratios, the epidemiologic predictors 'any history of contact with EVD case', and 'any contact with the healthcare setting' and the clinical predictors fatigue, sore throat, difficulty of swallowing, eye redness, confusion-disorientation, hematemesis, vomito-negro or black-vomiting, melena, gingival bleeding, and hemoptysis met the criteria to be included in multivariable analysis. After adjustment and shrinkage, for both the CPS and ECPS, predictors including fatigue, difficulty in swallowing, confusion-disorientation, red eyes, gingival bleeding, hematemesis, hemoptysis, and any history of contact with an EVD patient, had an adjusted likelihood ratio (aLR)≥1.5 or ≤0.67. Thus, they fulfilled the criteria to

**Table 1. Epidemiological and clinical characteristics of the included patients.**

| Predictors | Total number of suspects of EVD with the predictor n (%) | Total number of EVD patients with the predictor n (%) | OR (95% CI) |
|---|---|---|---|
| **Demographic and epidemiologic predictors** | | | |
| Age> = 25yrs | 4243 (40.8) | 401 (61.7) | 2.5 (2.1–2.9) |
| sex/female | 5037 (48.4) | 361 (55.8) | 1.4 (1.2–1.6) |
| High-risk profession (Health worker, Famer, Hunter) | 1936 (20.1) | 224 (39.6) | 2.6 (2.2–3.1) |
| Any history of contact with EVD-case | 1546 (15.1) | 415 (65.4) | 14.2 (12.0–17.0) |
| Traveled out of Butembo | 1416 (13.6) | 93 (14.4) | 1.0 (0.9–1.3) |
| Any history of contact with the healthcare setting (modern or traditional) | 615 (5.9) | 115(17.7) | 4.0 (3.2–5) |
| Contact with wildlife | 24 (0.2) | 2 (0.3) | 1.4 (0.3–5.9) |
| **Clinical predictors** | | | |
| **General symptoms** | | | |
| Fever | 3613 (34.7) | 295 (46.0) | 1.7 (1.4–1.9) |
| Malaise or fatigue | 5703 (54.7) | 499 (76.8) | 2.9 (2.4–3.5) |
| Headache | 5587 (53.6) | 311 (48.1) | 0.8 (0.7–0.9) |
| Loss of appetite | 4971 (47.4) | 393 (60.4) | 1.8 (1.5–2.1) |
| Nausea or vomiting | 3686 (35.4) | 325 (50.0) | 1.9 (1.6–2.2) |
| Abdominal pain | 4489 (43.1) | 262 (40.4) | 0.9 (0.8–1.0) |
| Non-bleeding diarrhea | 2752 (26.4) | 246 (37.8) | 1.8 (1.5–2.1) |
| Muscles or joint pain | 3498 (33.6) | 300 (46.5) | 1.8 (1.5–2.1) |
| Sore throat | 507 (4.9) | 81 (12.5) | 3.1 (2.4–4.0) |
| Skin rash | 296 (2.8) | 22 (3.4) | 1.2 (0.8–1.9) |
| Chest pain | 998 (9.6) | 98 (15.2) | 1.8 (1.4–2.2) |
| Difficulty in breathing | 574 (5.5) | 52 (8.0) | 1.5 (1.1–2.1) |
| Difficulty in swallowing | 879 (8.4) | 120 (18.5) | 2.7 (2.2–3.3) |
| Hiccups | 118 (1.1) | 13 (2.0) | 1.9 (1.0–3.4) |
| Red eyes | 377 (3.6) | 85 (13.1) | 4.9 (3.9–6.3) |
| Cough | 2605 (25.0) | 113 (17.4) | 0.6 (0.5–0.8) |
| **Haemorrhagic signs** | | | |
| Hematemesis | 97 (0.9) | 36 (5.5) | 9.3 (6.1–14.2) |
| Vomito negro | 22 (0.2) | 5 (0.8) | 4.4 (1.6–12.1) |
| Melena | 152 (1.5) | 20 (3.2) | 2.5 (1.5–3.9) |
| Gingival bleeding | 42 (0.4) | 29 (4.5) | 35.0 (18.1–67.6) |
| Epistaxis | 228 (2.2) | 19 (2.9) | 1.4 (0.9–2.2) |
| Petechiae | 4 (0.0) | 2 (0.3) | 15.0 (2–107.0) |
| Hemoptysis | 32 (0.3) | 5 (0.8) | 2.8 (1.1–7.3) |
| Hematuria | 44 (0.4) | 1 (0.2) | 0.3 (0.1–2.5) |
| Bleeding from any site | 764 (7.3) | 76 (11.7) | 1.7 (1.4–2.2) |
| **Neurologic signs** | | | |
| Confusion-disoriented | 120 (1.2) | 40 (6.2) | 7.9 (5.4–11.7) |
| **WHO case definition for the suspects** | 2707 (26.0) | 401(61.6) | 5.2 (4.4–6.1) |

OR: Crude odds ratio, 95%CI: 95% Confidence interval.

**Table 2. Accuracy of demographic, epidemiologic and clinical predictors for the diagnosis of EVD (compared to the reference standard laboratory confirmation).**

| Predictors | Sensitivity % (95%CI) | Specificity % (95%CI) | LR+ (95%CI) | LR– (95%CI) |
|---|---|---|---|---|
| **Demographic and epidemiologic predictors** | | | | |
| Age≥25 yrs. | 61.7 (57.8–65.4) | 60.6 (59.7–61.6) | 1.6 (1.4–1.7) | 0.6 (0.5–0.7) |
| sex/female | 55.8 (51.9–59.7) | 52.1 (51.1–53.1) | 1.2 (1.1–1.3) | 0.8 (0.7–0.9) |
| High-risk profession(Healthcare worker, Famer, Hunter) | 39.6 (35.5–43.8) | 79.9 (79.0–80.8) | 1.9 (1.7–2.3) | 0.8 (0.7–0.8) |
| Any history of contact with EVD-case | 65.4 (61.5–69.1) | 88.3 (87.6–88.9) | 5.6 (5–6.2) | 0.4 (0.3–0.4) |
| traveled out of Butembo | 14.4 (11.8–17.4) | 86.4 (85.7–87.1) | 1.1 (0.8–1.3) | 1.0 (0.9–1.0) |
| Any history of contact with the healthcare setting | 17.7 (14.9–20.9) | 94.9 (94.4–95.3) | 3.5 (2.7–4.4) | 0.9 (0.8–0.9) |
| Contact with wildlife | 0.3 (0.1–1.2) | 99.8 (99.7–99.9) | 1.5 (0.3–12.0) | 1.0 (0.9–1.0) |
| **General clinical predictors** | | | | |
| Fever or a history of fever | 46.0 (42.0–50.0) | 66.0 (65.1–67.0) | 1.4 (1.2–1.5) | 0.8 (0.7–0.9) |
| Malaise or fatigue | 76.8 (73.3–79.9) | 46.8 (45.8–47.8) | 1.4 (1.3–1.5) | 0.5 (0.4–0.6) |
| Headache | 48.1 (44.2–52.1) | 46.0 (45–47) | 0.9 (0.8–1.0) | 1.0 (1.0–1.0) |
| Loss of appetite | 60.4 (56.5–64.1) | 53.5 (52.5–54.5) | 1.3 (1.2–1.4) | 0.7 (0.7–0.8) |
| Nausea or vomiting | 50.0 (46.1–53.9) | 65.6 (64.7–66.6) | 1.5 (1.3–1.6) | 0.8 (0.7–0.8) |
| Abdominal pain | 40.4 (36.6–44.3) | 56.7 (55.7–57.7) | 0.9 (0.8–1.0) | 1.0 (0.9–1.0) |
| Non-bleeding diarrhea | 37.8 (34.1–41.7) | 74.4 (73.5–75.2) | 1.5 (1.3–1.7) | 0.8 (0.7–0.9) |
| Muscles or joints pain | 46.5 (42.6–50.4) | 67.3 (66.3–68.2) | 1.4 (1.3–1.6) | 0.8 (0.7–0.9) |
| Sore throat | 12.5 (10.1–15.4) | 95.6 (95.2–96.0) | 2.8 (2.1–3.9) | 0.9 (0.8–0.9) |
| Skin rash | 3.4 (2.2–5.2) | 97.2 (96.8–97.5) | 1.2 (0.7–2.1) | 1.0 (0.9–1.0) |
| Chest pain | 15.2 (12.6–18.3) | 90.8 (90.2–91.4) | 1.6 (1.3–2.1) | 0.9 (0.9–1.0) |
| Difficulty in breathing | 8.0 (6.1–10.4) | 94.7 (94.2–95.1) | 1.5 (1.1–2.1) | 1.0 (0.9–1.0) |
| Difficulty in swallowing | 18.5 (15.6–21.8) | 92.2 (91.7–92.8) | 2.4 (1.9–3.0) | 0.9 (0.8–0.9) |
| Hiccups | 2.0 (1.1–3.5) | 98.9 (98.7–97.3) | 1.8 (0.8–3.9) | 1.0 (0.9–1.0) |
| Red eyes | 13.1 (10.6–16.0) | 97.0 (96.7–97.3) | 4.4 (3.2–5.9) | 0.9 (0.8–0.9) |
| Cough | 17.4 (14.6–20.6) | 74.5 (73.6–75.4) | 0.7 (0.6–0.8) | 1.0 (1.0–1.0) |
| **Haemorrhagic clinical predictors** | | | | |
| Hematemesis | 5.5 (4–7.7) | 99.4 (99.2–99.5) | 9.2 (5.0–15.4) | 1.0 (0.9–1.0) |
| Vomito negro | 0.8 (0.3–1.9) | 99.8 (99.7–99.9) | 4.0 (1.0–19.0) | 1.0 (0.9–1.0) |
| Melena | 3.1 (2.1–5.0) | 98.7 (98.4–98.9) | 2.4 (1.3–4.5) | 1.0 (0.9–1.0) |
| Gingival bleeding | 4.5 (3.1–6.4) | 99.9 (99.8–99.9) | 45.0 (15.5–64) | 1.0 (0.9–1.0) |
| Epistaxis | 2.9 (1.8–4.6) | 97.9 (97.5–98.1) | 1.4 (0.7–2.4) | 1.0 (0.9–1.0) |
| Petechiae | 0.3 (0.1–1.2) | 100.0 (99.9–100.0) | ** | 1.0 (0.9–1.0) |
| Hemoptysis | 0.8 (0.3–1.9) | 99.7 (99.6–99.8) | 2.7 (0.8–9.5) | 1.0 (0.9–1.0) |
| Hematuria | 0.2 (0.0–1.0) | 99.6 (99.4–99.7) | 0.5 (0.0–3.3) | 1.0 (0.9–1.0) |
| Bleeding from any site | 11.7 (9.4–14.5) | 93.0 (92.4–93.5) | 1.7 (1.2–2.2) | 0.9 (0.9–1.0) |
| **Neuropsychiatric predictors** | | | | |
| Confusions or disorientation | 6.2 (4.5–8.4) | 99.2 (99.0–99.3) | 7.8 (4.5–12.0) | 0.9 (0.9–1.0) |
| **WHO's case definition for suspects** | 61.6 (57.7–65.3) | 76.4 (75.6–77.3) | 2.6 (2.4–2.9) | 0.5 (0.4–0.6) |

LR+: Positive likelihood ratio, LR–: Negative likelihood ratio and 95%CI: 95% confidence interval, **: + infinity.

be included in the scores. Table 3 presents the details of the results of the (E) CPS development analyses and the weighted scores produced after natural log-transformation.

The total score for each individual patient in the EVD surveillance dataset can range from -2 to +7 points for the CPS and from -4 to +9 points for the ECPS. Fig 2 and the S1 Table describe the distribution of the total of predictive scores CPS and ECPS in the EVD and non-EVD groups.

**Table 3. Crude and weighted score developed for clinical and extended clinical model score rules.**

| Predictors | Crude LR+/− | Clinical prediction model | | | Extended clinical prediction model | | |
|---|---|---|---|---|---|---|---|
| | | β-coefficient | adjusted LR+/− | Crude LR+/− | β-coefficient | adjusted LR+/− | Crude LR+/− |
| **Model intercept** | | -2.68 | | | -2.73 | | |
| **Risk exposures** | | | | | | | |
| Any history of patients contact | | n/a | | | 0.94 | | |
| Present | 5.60 | | n/a | n/a | | 5.04 | +2 |
| Absent | 0.40 | | n/a | n/a | | 0.41 | -1 |
| **Clinical signs/symptoms** | | | | | | | |
| **Fatigue/malaise** | | 1.03 | | | 1.15 | | |
| Present | 1.40 | | 1.46 | 0 | | 1.52 | 0 |
| Absent | 0.50 | | 0.49 | -1 | | 0.47 | -1 |
| **Dysphagia** | | 0.78 | | | 0.92 | | |
| Present | 2.4 | | 1.96 | +1 | | 2.20 | +1 |
| Absent | 0.90 | | 0.91 | 0 | | 0.89 | 0 |
| **Red eyes** | | 1.44 | | | 1.00 | | |
| Present | 4.40 | | 8.36 | +2 | | 4.37 | +1 |
| Absent | 0.90 | | 0.85 | 0 | | 0.90 | 0 |
| **Confusion** | | 0.80 | | | 1.09 | | |
| Present | 7.80 | | 5.17 | +2 | | 9.22 | +2 |
| Absent | 0.90 | | 0.96 | 0 | | 0.94 | 0 |
| **Gingival bleeding** | | 0.82 | | | 0.96 | | |
| Present | 45.00 | | 22.69 | +3 | | 38.42 | +4 |
| Absent | 1.00 | | 0.96 | 0 | | 0.96 | 0 |
| **Hematemesis** | | 1.07 | | | 1.42 | | |
| Present | 9.20 | | 10.73 | +2 | | 23.12 | +3 |
| Absent | 1.00 | | 0.95 | 0 | | 0.93 | 0 |
| **Hemoptysis** | | -0.68 | | | -1.59 | | |
| Present | 2.07 | | 0.51 | -1 | | 0.21 | -2 |
| Absent | 1.00 | | 1.00 | 0 | | 1.00 | 0 |

LR+/−: Positive/negative Likelihood Ratio; β-coefficient: Beta coefficient from multivariate analysis; n/a: Not applicable.

Fig 3 shows areas under the ROC curve (AUROC) plotting true-positive against the false-positive rate for each model of prediction, in the overall sample and in subgroups of suspects, depending or not if they fit with the WHO case definition. Applied to the whole dataset, the AUROC is 0.69 (95%CI: 0.67–0.71), 0.71(95%CI: 0.69–0.73) and 0.88 (95%CI: 0.86–0.89) for the WHO criteria, clinical and extended clinical model, respectively. Fig 3B and 3C show the AUROCs for the subgroups of EVD suspects (e.g. fitting or not the WHO criteria) (Fig 3). The AUROC for the extended score (ECPS) was significantly higher than the AUROC of the clinical prediction score (z = 12.1, p < 0.0001).

Fig 4 presents the calibration plots, which depict the observed proportion versus predicted probability of EVD plots for our scores. Generally, CPS well fitted the data and ECPS did not (Fig 4). Fig 5 and S2 Table present the results of the 10-fold cross-validation analysis for the two clinical prediction scores (CPS and ECPS). The mean cross-validation ROC (AUCCV) over the cross-validated samples with the ECPS was 0.87 (0.88 for the full dataset), and the mean AUCCV over the cross-validated samples with CPS was 0.71, the same as for the full dataset.

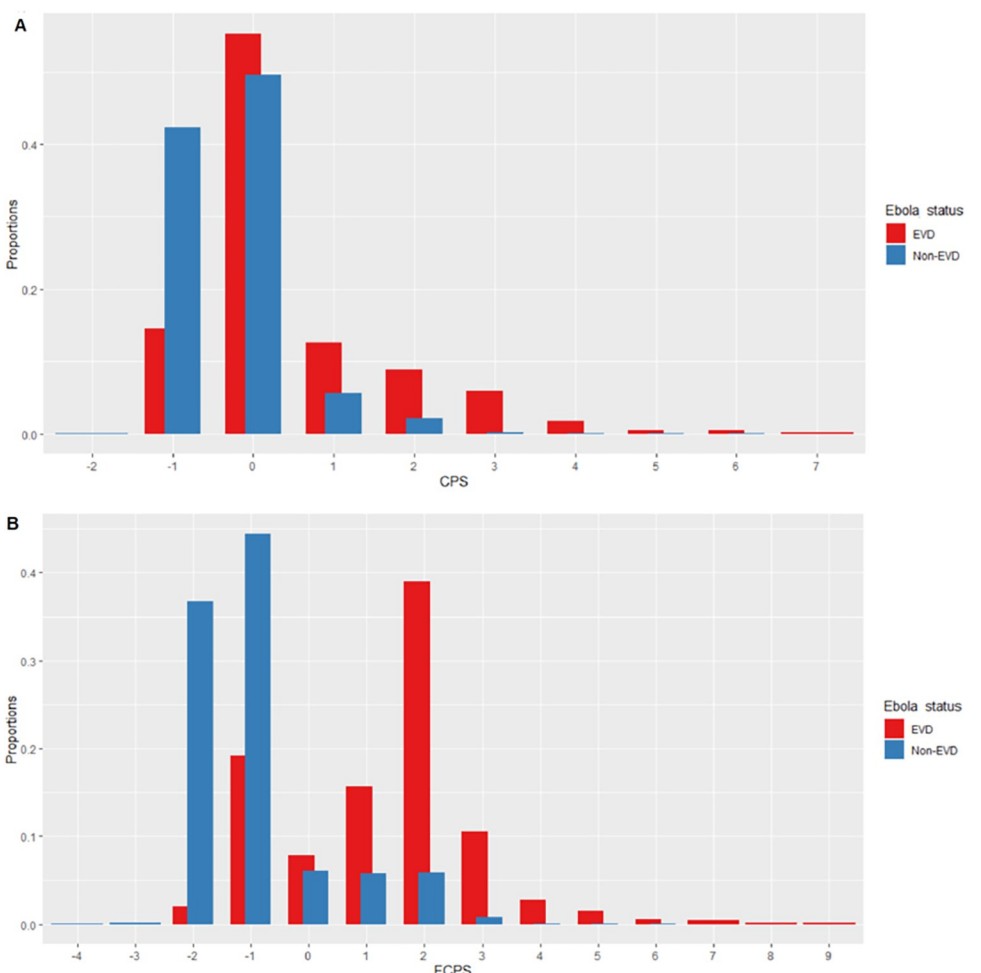

**Fig 2. Barplots depicting the distribution of the clinical prediction score and extended clinical prediction score in EVD and Non-EVD groups.**

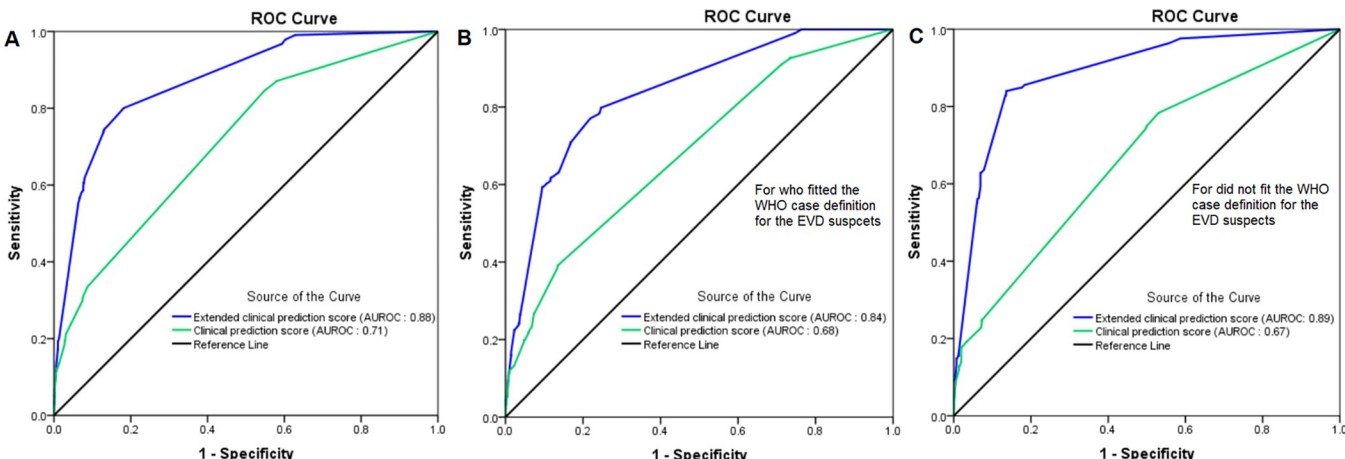

**Fig 3. Receiver operating curves (ROCs) plotting the discriminatory performance of the clinical prediction and extended clinical prediction scores for the screening of EVD.** (A) depicts the ROCs for the overall set of EVD suspects, (B) the ROCs for the set of EVD suspects who fit, and (C) presents the ROCs for the set of EVD suspects who do not fit with the WHO case definition used.

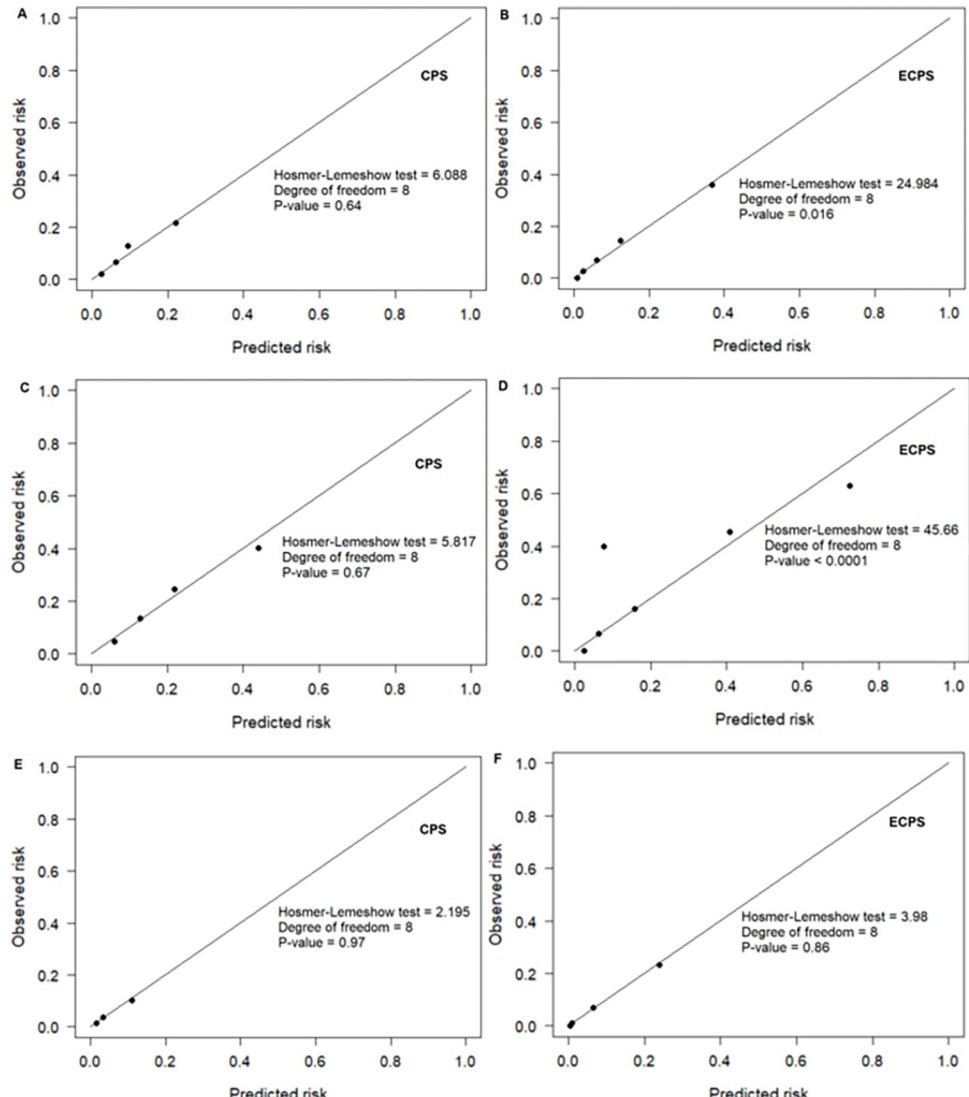

**Fig 4. Calibration plots for clinical prediction and extended clinical prediction scores for the screening of EVD.**
Calibration plots depict the observed proportion versus predicted probability of EVD. (A) and (B) present the
calibration plot for the overall set of EVD suspects, (C) and (D), the set of suspects who fit with the WHO case
definition, and (E) and (F) for those not fitting with the WHO case definition used.

At cut-off point value lower or equal to -1 for the clinical prediction score, chosen arbitrarily, sensitivity of 87.3% (95%CI: 84.4%–89.7%), and specificity of 42% (95%CI: 41%–43%) were estimated for the diagnosis of EVD. For the extended prediction score, the sensitivity of 81.4% (95%CI: 78.2% –84.3%), and specificity of 80.5% (95%CI: 79.7%–81.3%) were obtained at the cut-off point value of the score equal or lower than -1 for instance (**Table 4**).

Table 5 describes the performance of extended and clinical prediction scores on two subpopulations, one that fitted the WHO case definition for the suspect (with a disease prevalence of 14.8%) and the other that did not (with an EVD prevalence of 3.2%). At an arbitrarily chosen cut-off point value of -1 for the clinical prediction score, sensitivity of 91.0% (95% CI: 87.7%–93.6%) and specificity of 27.6% (95% CI: 25.8%–29.5%) were estimated in the 14.8% disease prevalence subgroup versus 76.4% (95% CI: 70.6%–81.5%) and 46.9% (95% CI: 45.7%–

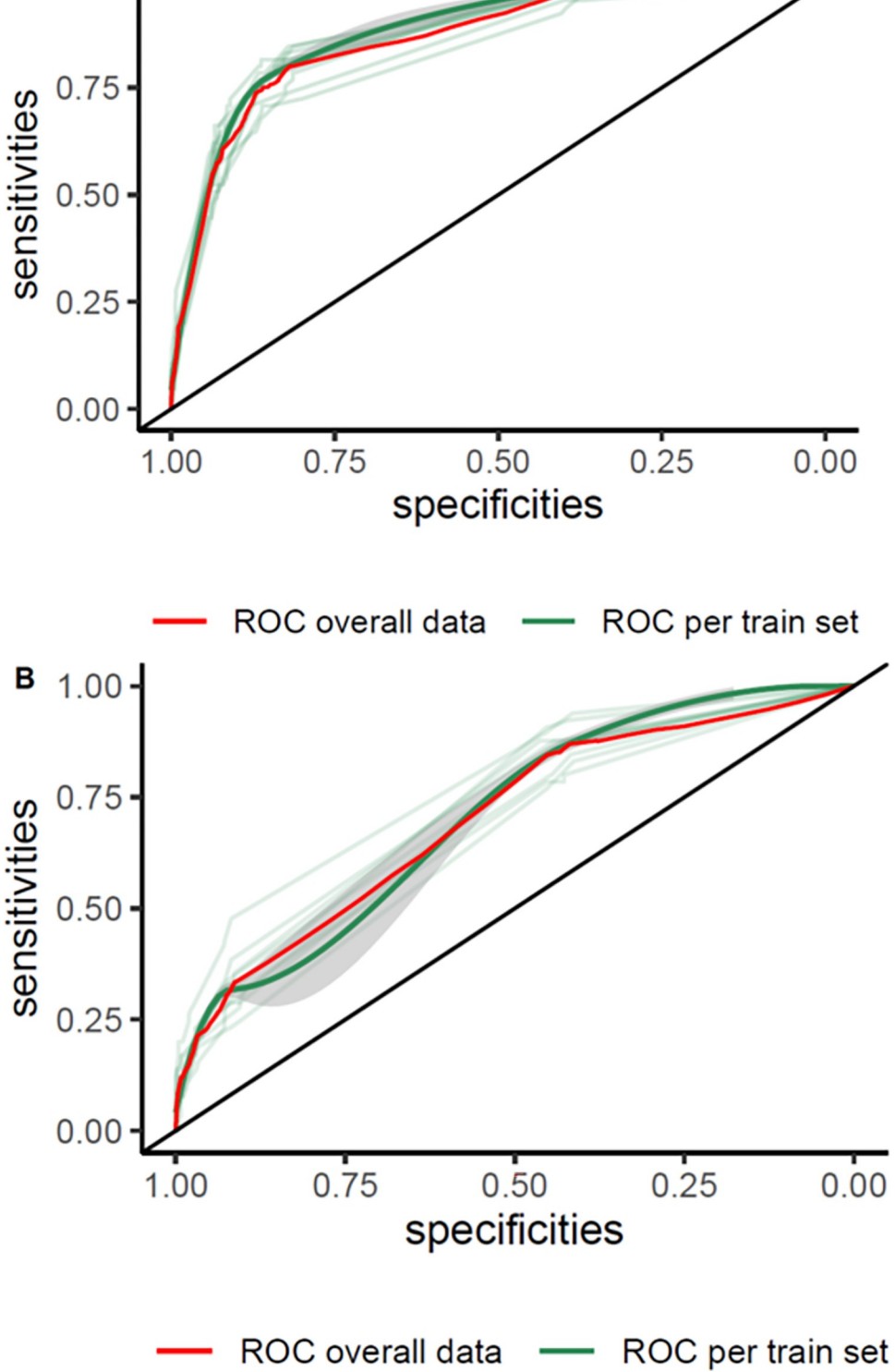

**Fig 5. Receiver operating characteristic (ROC) curves for the 10-fold cross validation of both Ebola clinical prediction scores on the test set.** (A) is the ROC for the extended clinical prediction score, (B) the ROC for the extended clinical prediction score.

**Table 4. Diagnostic performance at cut-points of total score for predicting EVD of clinical prediction score, extended clinical prediction score, and the WHO criteria for suspects.**

| Cut-off score/WHO criteria | Sensitivity % (95%CI) | Specificity % (95%CI) | PPV % (95%CI) | NPV % (95%CI) | AUROC (95%CI) |
|---|---|---|---|---|---|
| **WHO case definition for the suspect** | 61.6 (57.7–65.3) | 76.4 (75.6–77.3) | 14.8 (14.3–15.3) | 96.8 (96.4–97.2) | |
| **Clinical prediction score (CPS)** | | | | | 0.71 (0.69–0.73), a |
| ≥ -2 | 100.0 (100.0–100.0) | 0.0 (0.0–0.2) | 6.2 (5.9–6.7) | 100.0 (59–100.0) | |
| ≥ -1 | 85.4 (82.4–88.0) | 42.3 (41.4–43.3) | 9.0 (8.3–9.7) | 97.8 (97.2–98.2) | |
| ≥ 0 | 30.1 (26.6–33.8) | 91.9 (91.3–92.4) | 19.8 (17.4–22.2) | 95.2 (94.7–95.6) | |
| ≥ +1 | 17.5 (14.7–20.7) | 97.5 (97.2–97.8) | 31.8(27.1–37) | 94.7 (94.2–95.1) | |
| ≥ +2 | 8.6 (6.6–11.1) | 99.7 (99.5–99.8) | 62.9 (52.0–72.9) | 94.2 (93.9–94.7) | |
| ≥ +3 | 2.8 (1.7–4.4) | 99.9 (99.8–99.9) | 58.1(39.1–75.5) | 93.9 (93.4–94.4) | |
| ≥ +4 | 1.1 (0.5–2.3) | 100.0 (99.8–100.0) | 76.0 (52–90.6) | 93.8 (93.3–94.3) | |
| ≥ +5 | 0.6 (0.2–1.7) | 100.0 (100.0–100.0) | 80.0 (28.3–99.5) | 93.8 (93.3–94.3) | |
| ≥ +6 | 0.2 (0.01–1.0) | 100.0 (100.0–100.0) | 100.0(2.5–100.0) | 93.8 (93.3–94.2) | |
| ≥ +7 | 0.0 (0.0–0.6) | 100.0(100.0–100.0) | ** | 93.8 (93.3–94.2) | |
| **Extended clinical prediction score (ECPS)** | | | | | 0.88 (0.86–0.89), a |
| ≥ -4 | 100.0 (100.0–100.0) | 0.0 (0.0–0.2) | 6.2 (5.9–6.7) | 100.0 (59.0–100.0) | |
| ≥ -3 | 100.0 (100.0–100.0) | 0.2 (0.1–0.4) | 6.2 (5.8–6.7) | 100.0 (85.2–100.0) | |
| ≥ -2 | 98.0 (96.5–98.9) | 37.0 (36.0–37.9) | 9.4 (8.7–10.1) | 99.6 (99.3–99.8) | |
| ≥ -1 | 78.8 (75.4–81.8) | 81.4 (80.6–82.2) | 22.0 (20.4–23.8) | 98.3 (98.0–98.6) | |
| ≥ 0 | 71.0 (67.3–74.4) | 87.5 (86.8–88.1) | 27.4 (25.3–29.6) | 97.8 (97.5–98.1) | |
| ≥ +1 | 55.3 (51.4–59.2) | 93.2 (92.7–93.7) | 35.1 (32.2–38.1) | 96.9 (96.5–97.2) | |
| ≥ +2 | 16.3 (13.6–19.4) | 99.0 (98.8–99.2) | 52.7 (45.6–59.8) | 94.7 (94.2–95.1) | |
| ≥ +3 | 5.7 (4.1–7.8) | 99.8 (99.7–99.9) | 72.5 (58.2–84.1) | 94.1 (93.6–94.5) | |
| ≥ +4 | 2.9 (1.8–4.6) | 99.9 (99.8–100.0) | 76.0 (54.9–90.1) | 93.8 (93.3–94.3) | |
| ≥ +5 | 1.4(0.7–2.7) | 100.0 (99.9–100.0) | 81.8 (48.2–97.7) | 93.8 (93.3–94.3) | |
| ≥ +6 | 0.8 (0.3–1.9) | 100.0 (100.0–100.0) | 100.0 (47.8–100.0) | 93.8 (93.3–94.3) | |
| ≥ +7 | 0.3 (0.05–1.2) | 100.0 (100.0–100.0) | 100.0 (15.8–100.0) | 93.8 (93.3–94.2) | |
| ≥ +8 | 0.2 (0.1–1.0) | 100.0 (100.0–100.0) | 100.0 (2.5–100.0) | 93.8 (93.3–94.2) | |
| ≥ +9 | 0.0 (0.0–0.6) | 100.0 (100.0–100.0) | ** | 93.8 (93.3–94.2) | |

LR+: Positive likelihood ratio, LR−: Negative likelihood ration, PPV: Positive predictive value, NPV: Negative likelihood ratio, 95%CI: 95% Confidence Interval, AUROC: Area under ROC curve, **: + infinity, a: Comparison: AUROC for the extended clinical prediction score vs. the AUROC for the clinical prediction score (z = 12.1, p<0.0001).

48.0%) in the 3.2% prevalence subgroup. For the extended prediction score, the sensitivity of 76.6% (95% CI: 72.0%–80.6%) and specificity of 78.0% (95% CI: 76.2%–79.6%) in 14.8% subgroup suspects versus 82.4% (95% CI: 77.1%–86.9%) and 85.4% (95% CI: 81.6%–83.4%) in 3.2% prevalence were obtained at the cut-off point value of the score equal to or lower than -1, for instance (Table 5).

Fig 6 depicts the decision curves for isolating the suspects according to the prediction of the clinical and extended clinical scores compared to isolating all and isolating none of the suspects on a range of reasonable threshold probabilities. Fig 6A shows that the use of the clinical or extended clinical scores to isolate EVD-suspected patents offers higher net benefits than alternative reasonable strategies (e.g., isolating all or none of the suspects across all ranges of threshold probabilities of interest. When used to isolate EVD suspects, ECPS has more net benefit than any reasonable strategies on all ranges of probabilities below 70% and CPS has more net benefits than the "isolate all" or none suspects on the range of threshold probabilities between 2% and 67%.

**Table 5. Diagnostic performance of extended and clinical prediction scores for screening of Ebola suspects on two subgroups of suspects—those fitting or not fitting the WHO case definition.**

| Cut-off | Fitted with the WHO case definition used for the suspects | | | | Not fitted with the WHO case definition used for the suspects | | | |
|---|---|---|---|---|---|---|---|---|
| | (EVD prevalence: 14.8%, 401 EVD cases in 2707 suspects) | | | | (EVD prevalence: 3.2%, 250 EVD cases in 7725 suspects) | | | |
| | Sensitivity % (95% CI) | Specificity % (95% CI) | PPV % (95%CI) | NPV % (95%CI) | Sensitivity % (95% CI) | Specificity % (95% CI) | PPV % (95% CI) | NPV % (95% CI) |
| **Clinical prediction scores** | | | | | | | | |
| ≥ -2 | 100.0 (100.0–100.0) | 0.3 (0.1–0.7) | 14.9 (13.6–16.2) | 100.0 (100.0–100.0) | 100.0 (100.0–100.0) | 0.0 (0.0–0.0) | 3.2 (2.8–3.6) | ** |
| ≥ -1 | 91.0 (87.7–93.6) | 27.6 (25.8–29.5) | 17.9 (16.2–19.6) | 94.7 (93.0–96.4) | 76.4 (70.6–81.5) | 46.9 (45.7–48.0) | 4.6 (4.0–5.2) | 98.2 (97.8–98.6) |
| ≥ 0 | 35.2 (30.5–40.1) | 89.1 (87.7–90.3) | 35.9 (31.2–40.6) | 88.8 (87.5–90.1) | 22.0 (17.0–27.7) | 92.8 (92.2–93.3) | 9.2 (6.9–11.5) | 97.3 (96.9–97.7) |
| ≥ +1 | 19.2 (15.5–23.5) | 95.8 (94.9–96.6) | 44.3 (36.9–51.7) | 87.2 (85.8–88.6) | 14.8 (10.6–19.8) | 98.1 (97.7–98.2) | 20.2 (14.4–26.0) | 97.2 (96.8–97.6) |
| ≥ +2 | 10.0 (7.3–13.4) | 99.2 (98.7–98.7) | 68.9 (57.0–80.8) | 86.4 (85.1–87.7) | 6.4 (3.7–10.2) | 99.8 (99.7–99.9) | 51.6 (34.0–69.2) | 97.0 (96.6–97.4) |
| ≥ +3 | 3.5 (2.0–5.9) | 99.6 (99.2–99.8) | 60.9 (41.0–80.8) | 85.6 (84.3–86.9) | 1.6 (0.4–4.1) | 100.0 (100.0–100.0) | 50.0 (15.4–84.7) | 96.8 (96.4–97.2) |
| ≥ +4 | 1.8 (0.8–3.7) | 99.8 (99.5–99.9) | 58.3 (30.4–86.2) | 85.4 (84.1–86.7) | 0.0 (0.0–0.0) | 100.0 (100.0–100.0) | ** | 96.7 (96.3–97.1) |
| ≥ +5 | 1.0 (0.3–2.7) | 100.0 (97.7–100.0) | 80.0 (44.9–100.0) | 85.3 (84.0–86.6) | 0.0 (0.0–0.0) | 100.0 (100.0–100.0) | ** | 96.8 (96.4–97.2) |
| ≥ +6 | 0.3 (0.0–1.6) | 100.0 (100.0–100.0) | 100.0 (100.0–100.0) | 85.2 (83.9–86.5) | 0.0 (0.0–0.0) | 100.0 (100.0–100.0) | ** | 96.8 (96.4–97.2) |
| ≥ +7 | 0.0 (0.0–0.0) | 100.0 (100.0–100.0) | ** | 85.2 (83.9–86.5) | 0.0 (0.0–0.0) | 100.0 (100.0–100.0) | ** | 96.8 (96.4–97.2) |
| **Extended clinical prediction Scores** | | | | | | | | |
| ≥ -4 | 100.0 (100.0–100.0) | 0.3 (0.1–0.7) | 14.9 (13.6–16.2) | 100.0 (100.0–100.0) | 0.0 (0.0–0.0) | 0.0 (0.0–0.0) | 96.8 (96.4–97.2) | ** |
| ≥ -3 | 100.0 (100.0–100.0) | 1.0 (0.7–1.5) | 14.9 (13.6–16.2) | 100.0 (100.0–100.0) | 0.0 (0.0–0.0) | 0.0 (0.0–0.0) | 96.8 (96.2–97.4) | ** |
| ≥ -2 | 99.5 (98.0–99.9) | 24.0 (22.3–25.8) | 18.5 (16.9–20.1) | 99.6 (99.1–100.0) | 95.6 (92.3–97.8) | 41.0 (39.9–42.1) | 5.1 (4.5–5.7) | 99.6 (99.4–99.8) |
| ≥ -1 | 76.6 (72.0–80.6) | 78.0 (76.2–79.6) | 37.7 (34.4–41.0) | 95.0 (94.0–96.0) | 82.4 (77.1–86.9) | 85.4 (81.6–83.4) | 13.6 (11.9–15.3) | 99.3 (99.1–99.5) |
| ≥ 0 | 66.1 (61.2–70.7) | 87.6 (86.1–88.9) | 48.0 (43.8–52.2) | 93.7 (92.7–94.7) | 78.8 (73.2–83.7) | 87.5 (86.7–88.2) | 17.4 (15.2–19.6) | 99.2 (99.0–99.4) |
| ≥ +1 | 54.1 (49.1–59.1) | 91.7 (90.5–92.8) | 53.2 (48.4–58.0) | 92.0 (90.9–93.1) | 57.2 (50.8–63-4) | 93.7 (93.1–94.2) | 23.2 (19.9–26.5) | 98.5 (98.2–98.8) |
| ≥ +2 | 19.0 (15.3–23.2) | 98.1 (91.7–98.6) | 63.3 (54.7–71.9) | 87.4 (86.1–88.7) | 12.0 (8.216.7) | 99.3 (99.1–99.5) | 37.0 (26.5–47.5) | 97.1 (96.7–97.5) |
| ≥ +3 | 7.5 (5.2–10.6) | 99.5 (98.1–99.8) | 73.2 (59.6–86.8) | 86.1 (84.8–87.4) | 2.8 (1.1–5.7) | 100.0 (100.0–100.0) | 70.0 (41.6–98.4) | 96.9 (96.5–97.3) |
| ≥ +4 | 4.5 (2.8–7.1) | 99.8 (99.599.9) | 78.2 (61.3–95.1) | 85.7 (84.4–87.0) | 0.4 (0.0–2.2) | 100.0 (100.0–100.0) | 50.0 (0.0–100.0) | 96.8 (96.4–97.2) |
| ≥ +5 | 2.2 (1.1–4.4) | 99.9 (99.7–100) | 81.8 (59.0–100.0) | 85.5 (84.2–86.8) | 0.0 (0.0–0.0) | 100.0 (100.0–100.0) | ** | 96.8 (96.4–97.2) |
| ≥ +6 | 1.3 (0.5–3.1) | 100.0 (100.0–100.0) | 100.0 (100.0–100.0) | 85.3 (84.0–86.6) | 0.0 (0.0–0.0) | 100.0 (100.0–100.0) | ** | 96.8 (96.4–97.2) |
| ≥ +7 | 0.5 (0.1–2.0) | 100.0 (100.0–100.0) | 100.0 (100.0–100.0) | 85.3 (84.0–86.6) | 0.0 (0.0–0.0) | 100.0 (100.0–100.0) | ** | 96.8 (96.4–97.2) |
| ≥ +8 | 0.3 (0.0–1.6) | 100.0 (100.0–100.0) | 100.0 (100.0–100.0) | 85.2 (83.9–86.5) | 0.0 (0.0–0.0) | 100.0 (100.0–100.0) | ** | 96.8 (96.4–97.2) |

(*Continued*)

**Table 5.** (Continued)

| | Fitted with the WHO case definition used for the suspects | | | | Not fitted with the WHO case definition used for the suspects | | | |
|---|---|---|---|---|---|---|---|---|
| | (EVD prevalence: 14.8%, 401 EVD cases in 2707 suspects) | | | | (EVD prevalence: 3.2%, 250 EVD cases in 7725 suspects) | | | |
| Cut-off | Sensitivity % (95% CI) | Specificity % (95% CI) | PPV % (95%CI) | NPV % (95%CI) | Sensitivity % (95% CI) | Specificity % (95% CI) | PPV % (95% CI) | NPV % (95% CI) |
| ≥ +9 | 0.0 (0.0–0.0) | 100.0 (100.0–100.0) | ** | 85.1 (83.8–86.4) | 0.0 (0.0–0.0) | 100.0 (100.0–100.0) | ** | 96.8 (96.4–97.2) |

EVD: Ebola virus disease, PPV: Positive predictive value, NPV: Negative likelihood ratio, 95%CI: 95% Confidence Interval, **: + infinity.

In other words, irrespective of any differences in patient or health professional preferences to isolating the suspect, using (E) CPS improves the clinical decision-making (value) at the point of triage (e.g., increase isolation of EVD case and reduce isolation of false-positive EVD cases) on the ranges of threshold probabilities mentioned above. Additionally, the figure also shows that using the extended clinical prediction score has more net benefit than the clinical

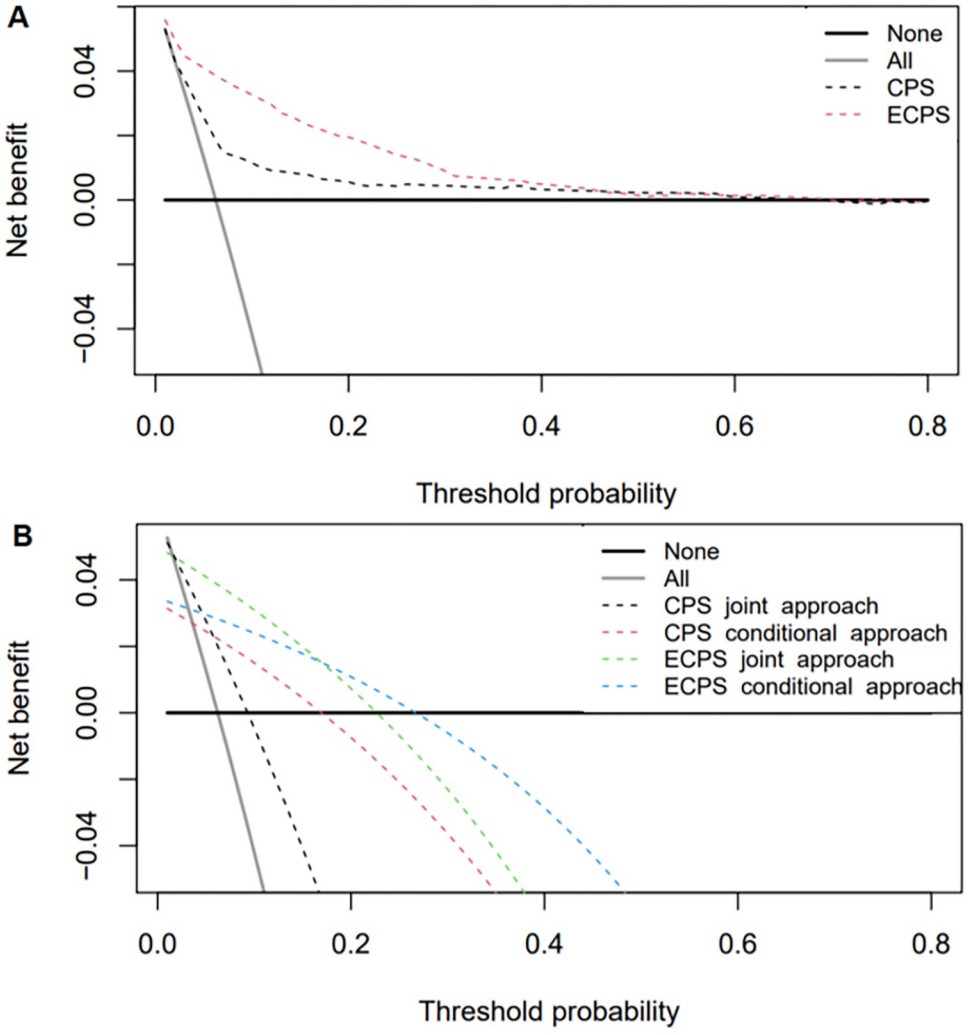

**Fig 6. The decision curve plotting the net benefit of the prediction scores at a range of threshold probability.** (A) represents the DCAs for the CPS and ECPS, and (B) the DCAs for the joint and conditional approaches with CPS and ECPS.

one at all ranges of probability thresholds. For instance, at a threshold probability of 5%, the net benefit of isolating the suspects using the prediction of the extended clinical score is higher than that of isolating the suspects according to the clinical prediction score, 0.041 and 0.026, respectively.

That means that, compared to the isolating "none of the suspects" strategy (by assuming that all suspects are negative), isolating the suspects on the basis of the CPS and ECPS is equivalent to strategies that correctly identify and isolate approximately 4.1 and 2.6 EVD cases per 100 suspects, respectively, without an increase in the number of non-EVD cases isolated as EVD cases. Furthermore, at this threshold, the net benefit of the CPS and ECPS is 0.013 and 0.028 greater than assuming all suspects are EVD. When these scores are used, there are 24.7% [= 0.013*100/5/(100–5)] fewer non-EVD cases classified and isolated as EVD cases with CPS and 53.2% [= 0.028*100/5/(100–5)] fewer non-EVD cases classified and isolated as EVD cases with EPCS per 100 suspects. These percentages represent the proportion of the reduction in isolation rate without missing any EVD cases (See **Table 6** and **Fig 6A**).

Both prediction score models add clinical value in the entire range of threshold probability up to 65%; over this threshold, they add no clinical benefit than isolating none of the suspects, as their net benefits are negative.

Finally, Fig 6B shows the DCAs for using the joint and conditional approaches with the two clinical prediction models. The figure shows that, relative to isolating all and isolating none, joint approaches for both ECPS and CPS had better patient clinical outcomes (net benefit) than their conditional approaches in a large range of threshold probabilities. It is between 2% and 9% range of threshold probabilities for the joint approach with CPS, 4% and 16% for the conditional approach with CPS, 2%, and 22% for the joint approach with ECPS, and 4% and 26% for the conditional approach with ECPS. This means that, for the preferences (or threshold probabilities) below this range of threshold probabilities, the isolate all suspect strategy is the best, and above this threshold, the best option is to isolate none of the suspects. Additionally, choosing a home for isolating suspects would be an option, for example.

## Discussion

### Main findings

We need sensitive and affordable diagnostic tools and efficient strategies to detect and control EVD outbreaks, especially in the settings with limited-resources and increased risk occurrence for EVD outbreaks in Africa. This study developed clinical prediction scores to improve the triage of suspect-cases at the point of-entry of a health facility during EVD outbreaks.

Importantly, the study has identified seven symptoms, including fatigue, difficulty in swallowing, red eyes, gingival bleeding, hematemesis, mental confusion, and hemoptysis, and one epidemiologic, having a history of any contact with an EVD case, as diagnostic predictors to build the two prediction scores (CPS and ECPS). These latter showed a better and excellent power in distinguishing the EVD from other conditions, respectively. The two models of scores are robust for the selected predictors. The addition of epidemiological factors clearly increased the diagnostic accuracy, and ECPS is preferred.

### Clinical symptoms and signs as well as risk exposure predictors

Our previous research on this data has compared the clinical and epidemiologic features of EVD and non-EVD to inform HCW on the case management decisions and infection control measures (Tshomba et al., in review). The study, along with others [31–33], identified and extensively discussed several epidemiological and clinical predictors associated with the diagnosis of EVD, most of which were also found in non-EVD patients during this outbreak.

**Table 6. Net benefit at some threshold probabilities and number of avoided unnecessary suspects isolation and/or further testing using the two developed models compared with isolating all suspects.**

| Threshold probability | Net Benefit | | | | | | | | Avoided unnecessary isolation and, or further testing per 100 EVD suspects | | | | | |
|---|---|---|---|---|---|---|---|---|---|---|---|---|---|---|
| | Via Isolate-all suspects strategy | Via isolate-none suspect strategy | Via using CPS to isolate | Via using ECPS to isolate | Via using the joint approach with CPS to isolate | Via using the conditional approach with CPS to isolate | Via using the joint approach with ECPS to isolate | Via using the conditional approach with ECPS to isolate | Using CPS to isolate | Using ECPS to isolate | Using the joint approach with CPS to isolate | Using the conditional approach with CPS to isolate | Using the joint approach with ECPS to isolate | Using the conditional approach with ECPS to isolate |
| 0.01 | 0.053 | 0.000 | 0.053 | 0.056 | 0.051 | 0.031 | 0.048 | 0.034 | 0.0 | 29.3 | -14.5 | -210.6 | -44.7 | -189.4 |
| 0.02 | 0.043 | 0.000 | 0.043 | 0.050 | 0.045 | 0.030 | 0.046 | 0.033 | 0.0 | 32.2 | 12.3 | -65.1 | 16.7 | -51.2 |
| 0.03 | 0.033 | 0.000 | 0.037 | 0.045 | 0.040 | 0.028 | 0.045 | 0.032 | 13.2 | 37.1 | 21.2 | -16.5 | 37.1 | -5.1 |
| 0.04 | 0.023 | 0.000 | 0.031 | 0.043 | 0.034 | 0.026 | 0.043 | 0.031 | 20.0 | 47.3 | 25.6 | 7.7 | 47.3 | 18.0 |
| 0.05 | 0.013 | 0.000 | 0.025 | 0.041 | 0.028 | 0.025 | 0.041 | 0.030 | 24.0 | 53.5 | 28.3 | 22.3 | 53.5 | 31.8 |
| 0.10 | -0.042 | 0.000 | 0.011 | 0.033 | -0.005 | 0.015 | 0.031 | 0.024 | 47.8 | 67.1 | 33.7 | 51.4 | 65.7 | 59.4 |
| 0.15 | -0.103 | 0.000 | 0.008 | 0.024 | -0.041 | 0.004 | 0.020 | 0.018 | 62.9 | 72.4 | 35.5 | 61.1 | 69.8 | 68.7 |
| 0.20 | -0.172 | 0.000 | 0.004 | 0.020 | -0.081 | -0.007 | 0.007 | 0.011 | 71.1 | 76.7 | 36.4 | 66.0 | 71.9 | 73.3 |
| 0.25 | -0.250 | 0.000 | 0.004 | 0.014 | -0.128 | -0.021 | -0.023 | 0.003 | 76.3 | 79.3 | 36.9 | 68.9 | 73.1 | 76.0 |
| 0.30 | -0.339 | 0.000 | 0.004 | 0.009 | -0.180 | -0.036 | -0.007 | -0.006 | 80.2 | 81.3 | 37.3 | 70.8 | 73.9 | 77.9 |
| 0.35 | -0.443 | 0.000 | 0.003 | 0.006 | -0.241 | -0.054 | -0.042 | -0.016 | 82.9 | 83.4 | 37.5 | 72.2 | 74.5 | 79.2 |
| 0.40 | -0.563 | 0.000 | 0.003 | 0.005 | -0.312 | -0.075 | -0.064 | -0.029 | 84.9 | 85.2 | 37.7 | 73.2 | 74.9 | 80.2 |
| 0.45 | -0.705 | 0.000 | 0.003 | 0.003 | -0.396 | -0.099 | -0.089 | -0.043 | 86.5 | 86.6 | 37.8 | 74.0 | 75.3 | 80.9 |
| 0.50 | -0.876 | 0.000 | 0.002 | 0.001 | -0.496 | -0.129 | -0.120 | -0.060 | 87.8 | 88.6 | 38.0 | 74.7 | 75.5 | 81.6 |
| 0.55 | -1.084 | 0.000 | 0.002 | 0.002 | -0.619 | -0.165 | -0.158 | -0.081 | 88.9 | 88.9 | 38.1 | 75.2 | 75.8 | 82.1 |
| 0.60 | -1.344 | 0.000 | 0.001 | 0.001 | -0.772 | -0.210 | -0.205 | -0.107 | 89.7 | 89.7 | 38.1 | 75.6 | 75.9 | 82.5 |
| 0.65 | -1.680 | 0.000 | 0.000 | 0.001 | -0.970 | -0.268 | -0.266 | -0.141 | 90.5 | 90.5 | 38.2 | 76.0 | 76.2 | 82.8 |
| 0.70 | -2.126 | 0.000 | -0.000 | 0.000 | -1.233 | -0.345 | -0.347 | -0.186 | 91.1 | 91.1 | 38.3 | 76.3 | 76.2 | 83.1 |
| 0.75 | -2.751 | 0.000 | -0.000 | 0.000 | -1.602 | -0.453 | -0.461 | -0.249 | 91.7 | 91.7 | 38.3 | 76.6 | 76.4 | 83.4 |
| 0.80 | -3.689 | 0.000 | -0.000 | -0.001 | -2.155 | -0.615 | -0.631 | -0.344 | 92.2 | 92.2 | 38.4 | 76.9 | 76.5 | 83.6 |

CPS: Clinical prediction score, ECPS: Extended clinical prediction score.

Importantly, the commonly reported risk factors and symptoms in EVD had previously been identified [34,35], and none of these clinical and epidemiologic predictors were found to be sensitive enough to better differentiate EVD cases. This study aimed to develop clinical prediction scores and evaluate their performance and clinical utility. Thus, our discussion will mainly focus on this aim.

## Clinical accuracy of symptoms, signs and risk exposure predictors

This study developed two prediction scores, which included fatigue, difficulty in swallowing, confusion-disorientation, red eyes, gingival bleeding, hematemesis, hemoptysis, and any history of contact with an EVD patient as a diagnostic predictor, some of which and their accuracy performance were consistent with those previously developed [21,22,36]. Although the methods used to develop prediction scores were different from ours, both models demonstrated better performance in discriminating EVD cases from others. Additionally, this discriminative performance remained the same, thus robust, in the two stratified sets of sample, e.g., for the set that fitted or did not fit the WHO case definition for the EVD suspects. Our study found that specificity (the ability of screening tests to recognize non-EVD suspects) for almost all cut-off points of the developed scores to be variable. Low specificity for a screening test implies that a high number of individuals will be isolated even though they are not EVD cases (false positives). Although the mathematical expressions of the basic accuracy metrics of the screening test do not depend on the prevalence, variations in specificity and sensitivity (the ability of screening tests to recognize EVD suspects) rely on the clinical variability in clinical presentations, which is the result of the variability in disease prevalence. Increasing the disease prevalence increases sensitivity but at the same time lowers specificity or vice versa. This was the case for the suspect sample we studied, in which the EVD prevalence was only 6.2%. For high disease prevalence suspects with many severe EVD patients, e.g., in the subgroup who fits the WHO case definition for the suspect (disease prevalence of 14.8%), the test sensitivity may appear better in this group of suspected populations than when applied to less severe EVD suspects. The increase in the prevalence of EVD decreases ruling out EVD cases (fewer false negatives). As a result, in clinical practice, health workers involved in the screening of EVD suspects should be aware of the accuracy variability related to EVD prevalence to make informed decisions. For this reason, using predictive values that translate the pre-test probability of disease to the post-test probability based on screening test results can be a better choice to guide clinicians in making rational screening and isolation decisions to control EVD. This post-probability depends additionally on the prevalence, or pre-test probability, of EVD—how common Ebola infections are in the EVD suspect group under consideration. In the same sense, the better the disease prevalence, the better the sensitivity of the screening test applied to discriminate high-risk suspects from low-risk ones.

The highest sum sensitivity-specificity is at -1 point and -1 point, respectively, for the clinical prediction score and the extended one; the models are associated with sensitivity of 85% and 42.30% and specificity of 78.8% and 81.40%. This means, for our prediction scores, the presence of a positive numeric score will be a useful argument for more investigations and/or the isolation of a suspect case. However, instead of choosing a sole cut-point, it is worth dividing it into three or four categories based on the numerical score, describing different probability levels of being useful in clinical practice. For instance, for these two models, one can consider a score of prediction equal or less than -2 as having a very low probability of the disease (i.e., less than 20%) and a score of prediction greater than +2 as having very high probability of the disease and act accordingly. With intermediate probabilities, e.g., between these two levels of predictive disease probabilities, one should consider the prediction score as not

conclusive and call for additional clinical evaluation. In this sense, using these models can allow the decision to be made on at least 40.7 percent (4141 non-EVD and 114 EVD cases) and 79.8% (7965 non-EVD and 360 EVD cases), respectively, for the clinical prediction score and the extended score.

Therefore, using these prediction scores can reduce the number of unnecessary isolations and/or additional procedures to be carried out for suspect-cases. The two prediction scores are associated with very low probability for EVD, and they avoid any delay in the isolation and case management for subjects with very high probabilities of disease in the isolation wards. Another way of using this score is to set up a threshold based on the greater sum sensitivity-specificity and then apply it to all suspect-cases. Those with scores above the threshold can be isolated, whereas subjects with a score under the threshold will not. However, the choice of the operational cut-off score should not only depend only on sensitivity and specificity but also on how professionals or patients weigh the errors of classification (false-positive and negative). Notably, the availability of human resources and health infrastructures and the context of use must be considered while managing the two prediction scores. For instance, in a context with adequate infrastructure, available effective drugs, a "disease-averse" context ("worried about the disease"), it is worth choosing a sensitive operational cut-off to catch a maximum number of EVD cases. A sensitive threshold can reduce the transmission in the community, and can be held to separate suspect-cases within the isolation wards to reduce cross-contamination between true-positive and false-positive cases. Conversely, in a context with poor health infrastructure and low density of the population, an "isolation-averse" context (e.g., "worried about the isolation" context), choosing a high specific cut-point (avoiding the FP) can reduce the risk of nosocomial infection among the patients with low-risk of contamination at the community level. With caution, prediction scores could be useful tools in the settings with limited-resources. The prediction scores do not intend to substitute the rational clinical or epidemiological judgments but to provide additional information, which can help healthcare workers to decide whether to isolate an EVD suspect case.

## Clinical usefulness of the developed models

Furthermore, we assessed the clinical usefulness of the two predictions scoring models developed by AUROC and the net benefit of using them to isolate EVD suspect-cases. We noted that both diagnostic prediction scores provided significantly higher AUROC and NB values relative to using clinically reasonable strategies (e.g., isolating all or none of the suspects). The AUROC is a summary measure of discrimination between individuals who experienced the disease and those who did not. Though it is subjective to determine the good value of the AUROC for assessing the disease risk, the AUROC values for the two scores developed are considered excellent, according to D'Agostino's AUROC classification [37]. Additionally, the two final scoring algorithms are quite robust for the selected variables. However, accuracy measures only show us about the models or tests' discriminative performance; they do not consider the clinical consequences of using these clinical prediction models to inform on their clinical usefulness in making decisions [29]. While addressing this issue, the study has also shown that incorporating prediction models into the decision-making to isolate suspect-cases would improve clinical results or outcomes linked to the use of WHO case definition for suspect-cases while awaiting the confirmation results. Their use avoids many unnecessary isolation and only a few true EVD cases could miss the isolation. Our findings demonstrate the utility of these models while identifying suspect-cases regardless of patient or professional preferences. The application of the DCA corresponds to the situation in which patients, EVD suspect-cases do not have a definite diagnosis yet. The healthcare workers or clinical team must

decide on whether the suspect-cases should be isolated or not, while the isolation exposes them to the cross-contamination among isolated people. This decision depends on the posterior probability of the disease for a suspect. Practically, the decision is straightforward if the suspected-case has very low or very high probability, e.g., less than 20% or greater than 80%, to have the disease. For intermediate probabilities, the decision is more complex and will require a consensus. The decision should consider the EVD's risk of spread-out (i.e., in communities or healthcare stings), disease severity and the context in which the decision is taken, including the availability of the resources, and should also be discussed on the basis of the perspective of the decision, which can be patient-centered, healthcare-centered, or societal-centered. Again, the DCAs represent rapid tools in the decision-making of healthcare workers during the surveillance and at the screening at the point of-entry since they do not require much additional data in the analysis. Nonetheless, they do not substitute traditional decision analysis, e.g., cost-effectiveness analysis, when their use is to inform and direct a policy choice.

### Strengths of developed predictive score systems

This study has explored the potential usefulness of clinical and epidemiological predictors in making clinical decisions in screening EVD suspect-cases with a large sample size. The usefulness of these predictive scores as rapid tests lies in their lack of cost and demand in terms of logistics Thus, standardized and applied screening tools for the health system frontline during outbreaks would be simple and beneficial in shortening the delay in decision-making by healthcare workers at the triage and/or case management.

### Limitations of the study

The main limitations of the study are First, the database used for this analysis contained many community cases from surveillance, which had missed clinical and epidemiologic data.

Second, the study included a population with a high prevalence of severe patients from the Ebola treatment unit during the epidemic period. Thus, the study missed most clinically asymptomatic or mild cases. Although the performance of our scores was excellent, their use during the inter-epidemic period or in other settings will need further evaluation.

Finally, the DCAs are simple and quick tools to improve the making of decisions, as their construction requires only one parameter, the range of threshold probabilities. However, for more complex decision choices, i.e., to inform policy, further research must capture the uncertainty involved before their use in a large context.

### In conclusion

An improved triage process and quick therapeutic decision could reduce the intra-hospital spread of infection and the disease-related mortality rate. This study developed diagnostic prediction scores that can aid in calculating the risk of being Ebola positive as a suspect case at the triage-point. These rapid and low-cost tools can help in decision-making to isolate EVD suspect cases at the triage point during an outbreak, as they constitute simple tools to assess the risk of having Ebola disease in suspected people. However, these tools still require external validation and a cost-effectiveness evaluation before being used on a large scale.

### Supporting information

**S1 Table. Distribution of clinical prediction score and extended clinical prediction score among suspect cases.**
(DOCX)

**S2 Table. 10-fold cross-validation accuracy for the clinical prediction score and extended clinical prediction score.**
(DOCX)

**S1 File. Notification chart used in the 2018–2020 DRC EVD outbreak.**
(PDF)

**S2 File. TRIPOD checklist for prediction model development and valuation.**
(DOCX)

**S3 File. North-Kivu/Butembo surveillance row data.**
(XLSX)

## Acknowledgments

The authors thank the local and international staff whose efforts contained the epidemic and collected data used in this study.

## Author Contributions

**Conceptualization:** Antoine Oloma Tshomba, Charles T. Kayembe, Dieudonné N. Mumba, Désiré D. Tshala-Katumbay, Sabue Mulangu.

**Data curation:** Antoine Oloma Tshomba, Daniel-Ricky Mukadi-Bamuleka, Olivier M. Tshiani, Richard O. Kitenge, Placide Mbala-Kingebeni.

**Formal analysis:** Antoine Oloma Tshomba, Bart K. M. Jacobs, Placide Mbala-Kingebeni.

**Funding acquisition:** Steve Ahuka-Mundeke, Désiré D. Tshala-Katumbay, Sabue Mulangu.

**Investigation:** Antoine Oloma Tshomba, Daniel-Ricky Mukadi-Bamuleka.

**Methodology:** Antoine Oloma Tshomba, Anja De Weggheleire, Charles T. Kayembe, Bart K. M. Jacobs, Lutgarde Lynen, Dieudonné N. Mumba, Désiré D. Tshala-Katumbay, Sabue Mulangu.

**Project administration:** Antoine Oloma Tshomba, Jean-Jacques Muyembe-Tamfum, Steve Ahuka-Mundeke, Dieudonné N. Mumba, Désiré D. Tshala-Katumbay, Sabue Mulangu.

**Supervision:** Charles T. Kayembe, Jean-Jacques Muyembe-Tamfum, Dieudonné N. Mumba, Désiré D. Tshala-Katumbay, Sabue Mulangu.

**Validation:** Daniel-Ricky Mukadi-Bamuleka, Anja De Weggheleire, Richard O. Kitenge, Bart K. M. Jacobs, Lutgarde Lynen, Placide Mbala-Kingebeni, Sabue Mulangu.

**Visualization:** Antoine Oloma Tshomba.

**Writing – original draft:** Antoine Oloma Tshomba, Daniel-Ricky Mukadi-Bamuleka, Anja De Weggheleire.

**Writing – review & editing:** Antoine Oloma Tshomba, Daniel-Ricky Mukadi-Bamuleka, Anja De Weggheleire, Olivier M. Tshiani, Richard O. Kitenge, Charles T. Kayembe, Bart K. M. Jacobs, Lutgarde Lynen, Placide Mbala-Kingebeni, Jean-Jacques Muyembe-Tamfum, Steve Ahuka-Mundeke, Dieudonné N. Mumba, Désiré D. Tshala-Katumbay, Sabue Mulangu.

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
