## [Decision Letter · Decision Letter 0]

5 Aug 2022

PONE-D-22-08999Development of Ebola Virus Disease prediction scores: screening tools for Ebola suspects at the triage-point during an outbreakPLOS ONE

Dear Dr. Tshomba,

Thank you for submitting your manuscript to PLOS ONE. After careful consideration, we feel that it has merit but does not fully meet PLOS ONE’s publication criteria as it currently stands. Therefore, we invite you to submit a revised version of the manuscript that addresses the points raised during the review process.

We look forward to receiving your revised manuscript.

Kind regards,

Jianhong Zhou

Staff Editor

PLOS ONE

https://journals.plos.org/plosone/s/file?id=ba62/PLOSOne_formatting_sample_title_authors_affiliations.pdf".

“This study has received partial support from NIH (Grant reference NIH FIC/R01EY031894).  The funder had no role in the design of the study; in the collection, analyses, or interpretation of data; in the writing of the manuscript, or in the decision to publish the results”

“DDK-T received a from US National Health Institute to support the neuro-Ebola project (Grant reference NIH FIC/R01EY031894)

NO - The funders had no role in study design, data collection and analysis, decision to publish, or preparation of the manuscript”

Additional Staff Editor Comments: We note that one or more reviewers has recommended that you cite specific previously published works. As always, we recommend that you please review and evaluate the requested works to determine whether they are relevant and should be cited. It is not a requirement to cite these works.

Reviewers' comments:

Reviewer's Responses to Questions

**Comments to the Author**

1. Is the manuscript technically sound, and do the data support the conclusions?

Reviewer #1: Yes

2. Has the statistical analysis been performed appropriately and rigorously? 

Reviewer #1: Yes

3. Have the authors made all data underlying the findings in their manuscript fully available?

Reviewer #1: Yes

4. Is the manuscript presented in an intelligible fashion and written in standard English?

Reviewer #1: Yes

5. Review Comments to the Author

Reviewer #1: The authors present an important and interesting manuscript that may be suitable for publication, pending revisions.

1. More effort should be directed at explaining the variations in specificity and their clinical impact when attempting to confirm the diagnosis of Ebola.

2. Mediocre to poor sensitivity means that as prevalence increases, Ebola is less likely to be ruled out. In other words, false omissions will increase as prevalence increases.

3. The authors should explicitly state the impact of prevalence in the abstract and devote some analysis of increasing prevalence in the body of the article.

4. The authors are encouraged to prove step-by-step that their definitions on lines 250-261 are correct and explain their derivations in detail.

3. The figures are a bit challenging. The legends could be improved. For example, which parts of the text represent the title and legend for Fig. 3? Did I miss something?

4. The column titles in Table 1 are not clear for the first 7 rows.

5. This statement in the Conclusion confuses, “…as they constitute the most straightforward tool to assess the risk while identifying suspect-cases with high probabilities of being Ebola positive….”. Do the authors mean that their approach will identify Ebola cases with high probability, or will a high pre-evaluation probability of Ebola be required for their approach to be successful?

6. The next statement in the Conclusion appears to be an unfounded claim, “The use of prediction scores reduces the number of unnecessary or at-risk isolation for non-EVD cases.” The concluding sentence in the abstract is more realistic, “However, these tools still require external validation and cost-effectiveness evaluation before any use at a large scale.” The authors should reconcile these two statements and eliminate the claim unless they can provide additional proof.

7. The intent of the authors is clear, and the approach has merit. However, this reviewer would be more comfortable using a high sensitivity and high specificity test to both rule out and rule in Ebola, in view of the mortality rates associated with this virus. Prevalence would have to be known to secure confidence in their approach. Too many healthcare workers, i.e., doctors and nurses, were lost during the West Africa outbreak to accept even modest risk of false omissions.

8. Perhaps the authors could construct a decision tree that would provide more insight into how they recommend triage decision makers use their prediction tool in real-world settings of different prevalence. At the same time, I suggest they review Bayesian math by consulting this recent paper and its full set of equations summarized in the supplement published in Diagnostics: https://www.mdpi.com/2075-4418/12/5/1216.

6. PLOS authors have the option to publish the peer review history of their article (what does this mean?). If published, this will include your full peer review and any attached files.

Reviewer #1: **Yes: **Gerald J. Kost, MD, PhD, MS, FAACC - Fulbright Scholar

---

## [Author Response · Author response to Decision Letter 0]

12 Sep 2022

Kinshasa, September 9, 2022

Tshomba Oloma Antoine

Institut National de Recherche Biomédicale (INRB) 

Kinshasa, Dem. Rep. of Congo, 

antotshomba@yahoo.fr

+243 815602451

September 9, 2022

Jianhong Zhou

Staff Editor

PLOS ONE journal

plosone@plos.org

Dear Editor Jianhong Zhou,

We are resubmitting our manuscript entitled “Development of Ebola Virus Disease prediction scores: screening tools for Ebola suspects at the triage-point during an outbreak” as a Research Article for consideration of publication in the PLOS ONE Journal. 

Before addressing all remarks, we want to express our gratefulness to the Editor and Reviewers’ for their overall very positive comments on our work and their suggestions for improvement. Through this letter, we have tried answering to the best of our knowledge the questions, suggestions and remarks provided by the Editors and Reviewers. 

None of the authors have a competing interest to declare and our manuscript has not been submitted, or accepted elsewhere. All authors have contributed to, seen, and approved the final, submitted version of the manuscript.

We have upload the following documents:

- The clean version of the Manuscript, file labeled "Manuscript"

- The track changes version of the manuscript, file labeled “Revised Manuscript with Track Changes”. 

- The covering letter addressing the editorial and referees’ comments, file labeled “Response to Reviewers”.

- We have provided through the system all relevant data (as DRC Ministry of Health since the outbreak has stopped and declared ended and data we provide are anonymized, no restriction still exists) as supporting information labeled “S3 File. North-Kivu/Butembo surveillance row data.” to the row surveillance data: https://docs.google.com/spreadsheets/d/1E1KSPO62siiCDVs3rpXq-J2OUqzNTZgq/edit?usp=sharing&ouid=117343120274024576555&rtpof=true&sd=true

Could The Plos one team download it, and upload on the submission System, on my behalf.

Additionally, for the funding statement, we dropped the paragraph related to “Funding information” in the text of the manuscript and keep this provided in the text of the manuscript. 

Moreover, on our behalf, we recommend the Editor to update the Funding statement as follows: 

“This study received partial a support from the Neuro-Ebola project, granted by the US National Institutes of Health (NIH). The funders played no role in study design, data collection and analysis, decision to publish, or preparation of the manuscript” 

Again thank you for all your deep and helpful remarks and recommendations you have addressed on our manuscript. 

We will look forward to hearing whether this manuscript can be considered of interest for publication in the PLOS ONE Journal and remain at your disposal for any required clarifications. 

Yours sincerely,

Antoine Tshomba

Date:Aug 05 2022 10:56AM

To:"Antoine Oloma Tshomba" antotshomba@yahoo.fr

From:"PLOS ONE" plosone@plos.org

Subject:PLOS ONE Decision: Revision required [PONE-D-22-08999]

PONE-D-22-08999

Development of Ebola Virus Disease prediction scores: screening tools for Ebola suspects at the triage-point during an outbreak

PLOS ONE

Dear Dr. Tshomba,

Thank you for submitting your manuscript to PLOS ONE. After careful consideration, we feel that it has merit but does not fully meet PLOS ONE’s publication criteria as it currently stands. Therefore, we invite you to submit a revised version of the manuscript that addresses the points raised during the review process.

We look forward to receiving your revised manuscript.

Kind regards,

Jianhong Zhou

Staff Editor

PLOS ONE

https://journals.plos.org/plosone/s/file?id=ba62/PLOSOne_formatting_sample_title_authors_affiliations.pdf".

Thank you very much for sending me these links. It was helpful in revising my manuscript. We re-organized our manuscript accordingly.

Not applicable to our study. The surveillance data we analyzed were collected under the emergence context and were anonymized. No consent was required. We stated that in the manuscript on lines 126–129.

“This study has received partial support from NIH (Grant reference NIH FIC/R01EY031894). The funder had no role in the design of the study; in the collection, analyses, or interpretation of data; in the writing of the manuscript, or in the decision to publish the results”

“DDK-T received a from US National Health Institute to support the neuro-Ebola project (Grant reference NIH FIC/R01EY031894)

NO - The funders had no role in study design, data collection and analysis, decision to publish, or preparation of the manuscript”

Indeed, we have dropped the paragraph related to “Funding information” in the manuscript text. 

We agree with you to publish the version in the funding statement section in the online submission. 

Please, on our behalf, update this statement as follows: 

“This study received partial support from the Neuro-Ebola project, granted by the US National Institutes of Health (NIH). The funders played no role in study design, data collection, and analysis, the decision to publish, or preparation of the manuscript.” 

Thank you very much. That is it! 

Since the North-Kivu and Ituri Ebola outbreaks in RDC have stopped and declared ended and the final report of the outbreak done, no restriction on the data exists. We will act with the PLOS ONE team to know how can we make these data available through the PLOS submission system. We wait to have advice from the team on this matter.

Therefore, we added this statement to the Data availability statement: 

“All relevant data are within the manuscript and its Supporting Information files.”

In addition, we added in supporting information the supplementary data file: S3 File. North-Kivu/Butembo surveillance row data. This link direct to the data: https://docs.google.com/spreadsheets/d/1E1KSPO62siiCDVs3rpXq-J2OUqzNTZgq/edit?usp=sharing&ouid=117343120274024576555&rtpof=true&sd=true

Additional Staff Editor Comments: We note that one or more reviewers has recommended that you cite specific previously published works. As always, we recommend that you please review and evaluate the requested works to determine whether they are relevant and should be cited. It is not a requirement to cite these works.

All relevant data and its Supporting Information files will be available within the paper through PLOS ONE system without any restriction.

Thank you very much for all the remarks and recommendations. They were of great input to improving the quality of this research. Again many thanks.

Reviewers' comments:

Reviewer's Responses to Questions

Comments to the Author

1. Is the manuscript technically sound, and do the data support the conclusions?

Reviewer #1: Yes

2. Has the statistical analysis been performed appropriately and rigorously?

Reviewer #1: Yes

3. Have the authors made all data underlying the findings in their manuscript fully available?

Reviewer #1: Yes

4. Is the manuscript presented in an intelligible fashion and written in standard English?

Reviewer #1: Yes

5. Review Comments to the Author

Reviewer #1: The authors present an important and interesting manuscript that may be suitable for publication, pending revisions.

1. More effort should be directed at explaining the variations in specificity and their clinical impact when attempting to confirm the diagnosis of Ebola.

Thank you very much. That was helpful to improve this manuscript. 

We added this statement on line 564-588

Our study found specificity (the ability of screening tests to recognize non-EVD suspects) for almost all cut-off points of the developed scores to be variable. Low specificity for a screening test implies that a high number of individuals will be isolated even though they are not EVD cases (false positives). Although the mathematical expressions of the basic accuracy metrics of the screening test do not depend on the prevalence, variations in specificity and sensitivity (the ability of screening tests to recognize EVD suspects) rely on the clinical variability in clinical presentations, which is the result of the variability in disease prevalence. Increasing the disease prevalence increases sensitivity but at the same time lowers specificity, or vice versa. This was the case for the suspect sample we studied, in which the EVD prevalence was only 6.2%. For high disease prevalence suspects with many severe EVD patients, e.g., in the subgroup who fit the WHO case definition for the suspect (disease prevalence of 14.8%), the test sensitivity may appear better in this group of suspected populations than when applied to less severe EVD suspects. The increase in the prevalence of EVD leads to a decrease in ruling out EVD cases (less false negatives). As a result, in clinical practice, health workers involved in the screening of EVD suspects should be aware of the accuracy variability related to EVD prevalence in order to make informed decisions. For this reason, using predictive values that translate the pre-test probability of disease to the post-test probability based on screening test results can be a better choice to guide clinicians in making rational screening and isolation decisions to control EVD. This post-probability depends additionally on the prevalence, or pre-test probability, of EVD—how common Ebola infections are in the EVD suspect group under consideration. In the same sense, the better the disease prevalence, the better the sensitivity of the screening test applied to discriminate high-risk suspects from low-risk ones.

2. Mediocre to poor sensitivity means that as prevalence increases, Ebola is less likely to be ruled out. In other words, false omissions will increase as prevalence increases.

In our response to last question (on lines 564-588), the response to this question is provided on lines 572-577 as follows:

Increasing the disease prevalence increases sensitivity but at the same time lowers specificity, or vice versa. This was the case for the suspect sample we studied, in which the EVD prevalence was only 6.2%. For high disease prevalence suspects with many severe EVD patients, e.g., in the subgroup who fit the WHO case definition for the suspect (disease prevalence of 14.8%), the test sensitivity may appear better in this group of suspected populations than when applied to less severe EVD suspects. The increase in the prevalence of EVD leads to a decrease in ruling out EVD cases (less false negatives)

 In addition, on lines 586-588 as follows: In the same sense, the better the disease prevalence, the better the sensitivity of the screening test applied to discriminate high-risk suspects from low-risk ones.

3. The authors should explicitly state the impact of prevalence in the abstract and devote some analysis of increasing prevalence in the body of the article.

Yes, we tried explaining the impact of disease prevalence on accuracy performance by describing our prediction scores’ diagnostic accuracy performance in two subgroup sets of data; one subgroup of suspects who fit the WHO case definition for the suspect and the other those who did not fit.

We added in the abstract part this statement: 

(On lines: 41-43): The EVD prevalence was 6.2% in the whole data set, 14.8% in the subgroup of suspects who fit the WHO Ebola case definition, and 3.2% for the suspects who did not fit this case definition. 

(On lines: 49-51): The diagnostic performance of the scores varied in the three disease contexts (the whole, fitting or not fitting the WHO case definition data sets).

We added in the method part of this statement (on lines 234–238)

Assuming that EVD prevalence in the two subgroups of suspects (e.g., those fitting or not fitting the WHO case definition) are different, we computed and described for each subgroup the diagnostic accuracy metrics to investigate the impact of the disease prevalence variation on the scores’ accuracy performance.

We have added the results part of this statement (on lines 333-335): 

The EVD prevalence among Ebola-suspected people was 6.2% in the total sample. The disease prevalence was 14.8% among suspects who fit the WHO case definition for the suspect (401 EVD cases in 2707 suspects) and 3.2% for those who did not fit (250 EVD cases in 7725 suspects).

Moreover, one additional table (on lines 523-526), Table 5. Diagnostic performance of extended and clinical prediction scores for screening of Ebola suspects on two subgroups of suspects; those fitting or not fitting the WHO case definition and we updated the table sequence in the text.

We have added the discussion part to this statement (on lines 564-588): 

Our study found specificity (the ability of screening tests to recognize non-EVD suspects) for almost all cut-off points of the developed scores to be variable. Low specificity for a screening test implies that a high number of individuals will be isolated even though they are not EVD cases (false positives). Although the mathematical expressions of the basic accuracy metrics of the screening test do not depend on the prevalence, variations in specificity and sensitivity (the ability of screening tests to recognize EVD suspects) rely on the clinical variability in clinical presentations, which is the result of the variability in disease prevalence. Increasing the disease prevalence increases sensitivity but at the same time lowers specificity or vice versa. This was the case for the suspect sample we studied, in which the EVD prevalence was only 6.2%. For high disease prevalence suspects with many severe EVD patients, e.g., in the subgroup who fit the WHO case definition for the suspect (disease prevalence of 14.8%), the test sensitivity may appear better in this group of suspected populations than when applied to less severe EVD suspects. The increase in the prevalence of EVD leads to a decrease in ruling out EVD cases (fewer false negatives). As a result, in clinical practice, health workers involved in the screening of EVD suspects should be aware of the accuracy variability related to EVD prevalence in order to make informed decisions. For this reason, using predictive values that translate the pre-test probability of disease to the post-test probability based on screening test results can be a better choice to guide clinicians in making rational screening and isolation decisions to control EVD. This post-probability depends additionally on the prevalence, or pre-test probability, of EVD—how common Ebola infections are in the EVD suspect group under consideration. In the same sense, the better the disease prevalence, the better the sensitivity of the screening test applied to discriminate high-risk suspects from low-risk ones.

4. The authors are encouraged to prove step-by-step that their definitions on lines 250-261 are correct and explain their derivations in detail.

Indeed! We added the followed statement on lines 269-285:

For more detail on the technical approach of our derivation, please consider the paper by Vickers (http://www.decisioncurveanalysis.org/). In practice, implementing the decision curve analysis requires the following height steps:

1. Choose a value for pt, e.g., the threshold probability, a composite of patient and health workers’ preferences (a theoretical risk level where the expected benefit of isolating a suspect is equal to the expected benefit of avoiding it).

2. With this threshold probability as a cut-point, calculate the number of true positives and false positives using 2x2 tables that summarize the positive or negative results.

3. Compute the net benefit of the prediction model.

4. Repeat steps 2–3 by varying pt over a range of interest.

5. Plot net benefit on the y-axis against pt on the x-axis to draw the decision curve for the model.

6. Repeat steps 1–5 for every model to compare.

7. Repeat steps 1–5 for the strategy that assumes all EVD suspects are positive.

8. Finally, a straight line drawn parallel to the x-axis at y=0 represents the net benefit associated with the strategy of assuming that all suspects are negative.

3. The figures are a bit challenging. The legends could be improved. For example, which parts of the text represent the title and legend for Fig. 3? Did I miss something?

Thank you for your helpful remarks. We tried to make them clearer. The bold part of writing is the titles and this not bold is legends.

We re-wrote figure titles and legends as follows:

Fig 3. Receiver operating curves (ROCs) plotting the discriminatory performance of the clinical prediction and extended clinical prediction scores for the screening of EVD. 

A depicts the ROCs for the overall set of EVD suspects, B is the ROCs for the set of EVD suspects who fitted, and C presents the ROCs for the set of EVD who did not fit with the WHO case definition used.

Fig 4. Calibration plots for clinical prediction and extended clinical prediction scores for the screening of EVD. 

Calibration plots depict the observed proportion versus the predicted probability of EVD. A and B represent the calibration plot for the overall set of EVD suspects, C and D graphs depict the calibration for the set of suspects who fitted, and E and F are the calibration plots for those who did not fit with the WHO case definition used.

Fig 5. Receiver operating characteristic (ROC) curves for the 10-fold cross-validation of both Ebola clinical prediction scores on the test set. 

A is the ROC for the extended clinical prediction score, B is the ROC for the extended clinical prediction score

Fig 6. Decision curve plotting net benefit of the prediction scores at a range of threshold probability. 

A represents the DCAs for the CPS, ECPS, and B, the DCAs for the joint and conditional approaches with CPS and ECPS

4. The column titles in Table 1 are not clear for the first 7 rows.

Thank you very much. Indeed, we replaced the word symptom with predictor in the column titles. Therefore, it can work with all rows. 

Column names are changed now: Total number of suspects of EVD with the symptom n (%) to Total number of suspects of EVD with the predictor n (%) and Total number of EVD patients with the symptom n (%) to Total number of EVD patients with the predictor n (%)

5. This statement in the Conclusion confuses, “…as they constitute the most straightforward tool to assess the risk while identifying suspect-cases with high probabilities of being Ebola positive….”. Do the authors mean that their approach will identify Ebola cases with high probability, or will a high pre-evaluation probability of Ebola be required for their approach to be successful?

Thank you for your remark.

We re-wrote the conclusion (on lines as follows:

An improved triage process and quick therapeutic decisions could reduce the intra-hospital spread of infection and the disease-related mortality rate. This study developed diagnostic prediction scores can help to calculate the risk of being Ebola positive as a suspect case at the triage point. These rapid and low-cost tools can help in decision-making to isolate EVD suspect cases at the triage point during an outbreak, as they constitute simple tools to assess the risk of having Ebola disease in suspected people. However, these tools still require external validation and a cost-effectiveness evaluation before being used on a large scale.

6. The next statement in the Conclusion appears to be an unfounded claim, “The use of prediction scores reduces the number of unnecessary or at-risk isolation for non-EVD cases.” The concluding sentence in the abstract is more realistic, “However, these tools still require external validation and cost-effectiveness evaluation before any use at a large scale.” The authors should reconcile these two statements and eliminate the claim unless they can provide additional proof.

Thank you for the remarks as for the previous last response. That were helpful.

We rewrote the whole conclusion as above: 

An improved triage process and quick therapeutic decisions could reduce the intra-hospital spread of infection and the disease-related mortality rate. This study developed diagnostic prediction scores can to calculate the risk of being Ebola positive as a suspect case at the triage point. These rapid and low-cost tools can help in decision-making to isolate EVD suspect cases at the triage point during an outbreak, as they constitute simple tools to assess the risk of having Ebola disease in suspected people. However, these tools still require external validation and a cost-effectiveness evaluation before being used on a large scale.

7. The intent of the authors is clear, and the approach has merit. However, this reviewer would be more comfortable using a high sensitivity and high specificity test to both rule out and rule in Ebola, in view of the mortality rates associated with this virus. Prevalence would have to be known to secure confidence in their approach. Too many healthcare workers, i.e., doctors and nurses, were lost during the West Africa outbreak to accept even modest risk of false omissions.

Thank you very much for these remarks. 

We added the information related to the disease prevalence in the whole sample and two stratified samples according to their fitting with the WHO case definition for the suspect. 

We added on lines 333-335 this statement: 

The EVD prevalence among Ebola-suspected people was 6.2% in the total sample. The disease prevalence was 14.8% among suspects who fit the WHO case definition for the suspect (401 EVD cases in 2707 suspects) and 3.2% for those who did not fit (250 EVD cases in 7725 suspects).

8. Perhaps the authors could construct a decision tree that would provide more insight into how they recommend triage decision makers use their prediction tool in real-world settings of different prevalence. At the same time, I suggest they review Bayesian math by consulting this recent paper and its full set of equations summarized in the supplement published in Diagnostics: https://www.mdpi.com/2075-4418/12/5/1216.

We thank you very much for this recommendation. In this manuscript, mainly, we have focused on the prediction score development and show how beneficial their implementation can be. We are on the next focused on cost-effectiveness analysis. In this, we have used our scores and applied them to real-world settings through a decision analytic frame. Both the Bayesian approach and sensitivity analysis will be included in the analyses. 

6. PLOS authors have the option to publish the peer review history of their article (what does this mean?). If published, this will include your full peer review and any attached files.

Do you want your identity to be public for this peer review? For information about this choice, including consent withdrawal, please see our Privacy Policy.

Reviewer #1: Yes: Gerald J. Kost, MD, PhD, MS, FAACC - Fulbright Scholar

Thank you. Yes, we uploaded them here.

---

## [Editor Report · Decision Letter 1]

22 Nov 2022

Development of Ebola Virus Disease prediction scores: screening tools for Ebola suspects at the triage-point during an outbreak

PONE-D-22-08999R1

Dear Dr. Oloma Tshomba,

We’re pleased to inform you that your manuscript has been judged scientifically suitable for publication and will be formally accepted for publication once it meets all outstanding technical requirements.

Kind regards,

M. Kariuki Njenga

Academic Editor

PLOS ONE
---

## [Editor Report · Acceptance letter]

8 Dec 2022

PONE-D-22-08999R1 

Development of Ebola Virus Disease prediction scores: screening tools for Ebola suspects at the triage-point during an outbreak 

Dear Dr. Tshomba:

I'm pleased to inform you that your manuscript has been deemed suitable for publication in PLOS ONE. Congratulations! Your manuscript is now with our production department. 

Kind regards, 

on behalf of

Dr. M. Kariuki Njenga 

Academic Editor

PLOS ONE